# Response surface methodology reveals proportionality effects of plant species in conservation plantings on occurrence of generalist predatory arthropods

Joseph M. Patt [ID]*, Aleena M. Tarshis Moreno, Randall P. Niedz

U.S. Horticultural Research Laboratory, USDA-Agricultural Research Service, Fort Pierce,Florida, United States of America

* joseph.patt@usda.gov

**Data Availability Statement:** All relevant data are within the manuscript and its Supporting Information files.

## Abstract

Multivariate geometric designs for mixture experiments and response surface methodology (RSM) were tested as a means of optimizing plant mixtures to support generalist predatory arthropods. The mixture design included 14 treatment groups, each comprised of six planters and having a proportion of 0.00, 0.17, 0.33, 0.66, or 1.00 of each plant species. The response variable was the frequency of predators trapped on sticky card traps placed in each group and replaced 2 times per week. The following plant species were used: Spring 2017: *Euphorbia milii*, *E. heterophylla*, and *Phaseolus lunatus*; Summer 2017: *E. milii*, *Fagopyrum esculentum*, and *Chamaecrista fasciculata*; and, Summer 2018: *E. milii*, *F. esculentum*, and *Portulaca umbraticola*. Predator occurrence was influenced by: 1) Linear mixture effects, which indicated that predator occurrence was driven by the amount of a single plant species in the mixture; or, 2) Nonlinear blending effects, which indicated that the plant mixture itself had emergent properties that contributed to predator occurrence. Predator abundance was highest in the Spring 2017 experiment and both linear mixture effects and nonlinear blending effects were observed. Predator occurrence decreased in subsequent experiments, which were conducted in the warmer summer months. In both Summer experiments, only linear mixture effects were observed, indicating that predator occurrence was driven by the amount of a single plant species in the test mixtures: *Euphorbia milii* in 2017 and *Portulaca umbraticola* in 2018. The results showed that not only did the species composition of a plant mixture drive predator occurrence but that proportionality of species contributed to the outcome as well. This suggests that, when formulating a plant mixture to aid in conservation biological control consideration should be given to the proportion of each plant species included in the mixture. RSM can be an important tool for achieving the goal of optimizing mixtures of plants for conservation biological control.

**Funding:** This research was funded by the USDA Agricultural Research Service. The funders had no role in study design, data collection and analysis, decision to publish, or preparation of the manuscript.

**Competing interests:** The authors have declared that no competing interests exist.

## Introduction

The addition of certain plant species to depauperate landscapes can provide significant resource subsidies to predaceous and parasitic arthropods, and may result in increased suppression of pest insects [1–8]. This is because development, survivorship, fecundity, and foraging efficiency of natural enemies is enhanced by the presence of nutritional resources such as nectar, pollen or alternate prey or hosts, provided by certain non-crop plants [1,7, 9–11], which we will refer to as 'conservation' plants. Within monoculture agroecosystems, nutritional resource subsidies from conservation plants can be especially critical to natural enemies during periods of prey scarcity [12–17] or to provide refuge when crops are treated with insecticides [18]. The incorporation of conservation plants to support natural enemies in agro-ecosystems is a component of a broader approach called conservation biological control, a management strategy which seeks to increase the abundance and diversity of arthropod biological control agents to improve pest control.

Biological control agents typically lack the specialized mouthparts and foraging ability needed to obtain nectar or pollen in deep corollas or hidden by other floral structures [19–21]. Therefore, an important consideration for the IPM practitioner is to select conservation plant species whose floral morphology is compatible with the mouthpart morphologies and foraging behaviors of target natural enemies or that have extrafloral nectaries, which are exposed and accessible [22,23]. Campbell et al. [24] illustrated the importance of using a floral-trait based approach by showing that plots with flowers having exposed nectaries, such as coriander (*Coriandrum sativum* L), attracted significantly more hoverflies and parasitoid wasps, whereas plots with flowers having nectaries enclosed by floral structures, such as birdsfoot trefoil (*Lotus corniculatus* L.), attracted few natural enemies but were extensively visited by bumblebees, which are adept at foraging from such flowers.

Incorporating monoculture strips of conservation plants within or around the cropping system can effectively attract natural enemies and lead to the suppression of certain crop pests. For example, strips of flowering alyssum (*Lobularia maritima* L) support parasitoids of lettuce aphids [25–27] while the predators of rice hoppers are supported by the extrafloral nectaries of sesame planted on berms surrounding rice paddies [28]. While monoculture strips provide flexibility with respect to crop rotation and other agronomic considerations, it may be desirable to provide polycultures of conservation plants in particular agroecosystems. Reasons for doing so would include ensuring a continuous supply of nutritional resources for natural enemies over an extended period of time, providing multiple trophic resources, and providing shelter and other microhabitat resources [7, 29–31].

Wäckers and van Rijn [32] summarized the important features needed for selecting conservation plants for the purpose of providing nutritional resources for particular natural enemy taxa. Still, the literature on the efficacy of conservation plant mixtures on enhancing conservation biological control shows that the interactive effects between plant species and the insect taxa they support is complex, and more work is needed to optimize plantings for this purpose [33–36]. The effectiveness of a given plant mixture is, in large part, a function of its plant species composition and its functional diversity; i.e., the functional traits of the plants [37–41]. However, the proportionality of plant species in a mixture may drive its functionality as well; this has received little attention in conservation biological control.

We are developing a conservation biological control strategy for *Diaphorina citri* Kuwayama (Hemiptera: Liviidae) in Florida. This psyllid vectors the causal agent of Huanglongbing (HLB) disease of citrus, also known as citrus greening [42–44], a highly destructive disease that threatens citrus production worldwide [45,46]. In Florida, HLB has severely impacted citrus fruit production since it was first detected in 2005 [47,48]. The presumed causal agent of HLB is the phloem-

limited, gram positive bacterium *Candidatus* Liberibacter asiaticus (*C*Las) [45]. *Diaphorina citri* is a specialist herbivore that feeds on the phloem of *Citrus* and its close relatives [42–44]. The psyllid moves between citrus trees and ornamental shrubs, such as *Murraya paniculata* (L) Jack growing in residential areas and urban landscapes and commercial citrus groves, spreading HLB from area to area [49–51]. Efforts to establish an exotic parasitic wasp (*Tamarixia radiata* Waterson) that attacks immature psyllids have had mixed results [52–54], and control measures for psyllids dwelling in residential and urban areas are essentially non-existent.

Although the psyllid is vulnerable to attack by generalist predators [55–57], commercial citrus growers rely almost exclusively on chemical controls to suppress psyllid populations [58,59]. A strong reliance of insecticides has failed to prevent the spread of HLB in Florida, where nearly 100% of commercial citrus groves are infected with HLB, and *D. citri* populations with resistance to multiple insecticide classes have emerged there and elsewhere [60–63]. Rather than depend on insecticidal control of *D. citri*, commercial citrus growers in Florida are opting to purchase other measures such as antimicrobial applications to suppress *C*Las [64,65] and enhanced fertilization and foliar nutritional sprays to enhance tree health. The intensive dependence on chemical control has adversely affected populations of natural enemies residing in or near citrus groves [56,66,67]. As chemical suppression is reduced, natural enemies are expected to return to citrus groves [68], and growers are beginning to consider biological control as an option to cut costs. A recent analysis indicated that commercial citrus growers in Florida could make significant economic savings by applying insecticides only during a few periods when they are most effective [68]. This approach would make insecticide applications more effective; and, the reduction in spray frequency would permit natural enemies to enter the groves and provide a high level of *D. citri* suppression. Clearly there is a growing need to develop and implement effective conservation biological control strategies to suppress *D. citri* populations in commercial citrus groves and nearby residential and commercial landscapes.

The effective implementation of conservation biological control requires detailed knowledge of the agroecosystem [69]. As a first step in developing a conservation biological control strategy for *D. citri*, we compared the efficacy of single-species versus mixed species plantings with respect to supporting the psyllid's predators. Multivariate mixture geometric designs and response surface methodology (RSM) is an effective statistical tool for evaluating the role of proportionality of individual components in the properties of mixtures [70]. Here, we tested whether RSM could be used to optimize mixtures of three conservation plant species with respect to their ability to provide functional habitat for the natural enemies of *D. citri*. We felt that by approaching optimization of conservation plantings as a mixture problem, we could determine the effects of the proportions of different plant species on natural enemy occurrence. Mixture designs can identify the driver components in complex mixtures of substances ranging from tissue culture components to insect diets to mating disruption pheromones [71]. Importantly, a mixture design can detect and quantify proportionality effects, which manifest as 'nonlinear blending' (sometimes called quadratic blending coefficients) of mixture components. Nonlinear blending occurs when the proportionality of components matters. In other words, when the response is greater or lesser than the sum of the components would otherwise indicate, then the mixture is exhibiting nonlinear blending effects. Because factorial-based designs confound the effects of proportionality with the effects of amount, they cannot determine the effects of proportionality. Only a mixture design can determine these effects. Varying multiple components simultaneously identifies which components are extraneous and which are vital in identifying an efficient mixture. A detailed description of these statistical methods is provided by Niedz and Evens [71] and their application in biological systems by Niedz and Marutani-Hert [72]. Examples of entomological studies which used RSM are provided by Lapointe et al. [73] and Pascacio-Villafán et al. [74] for determining essential ingredients in

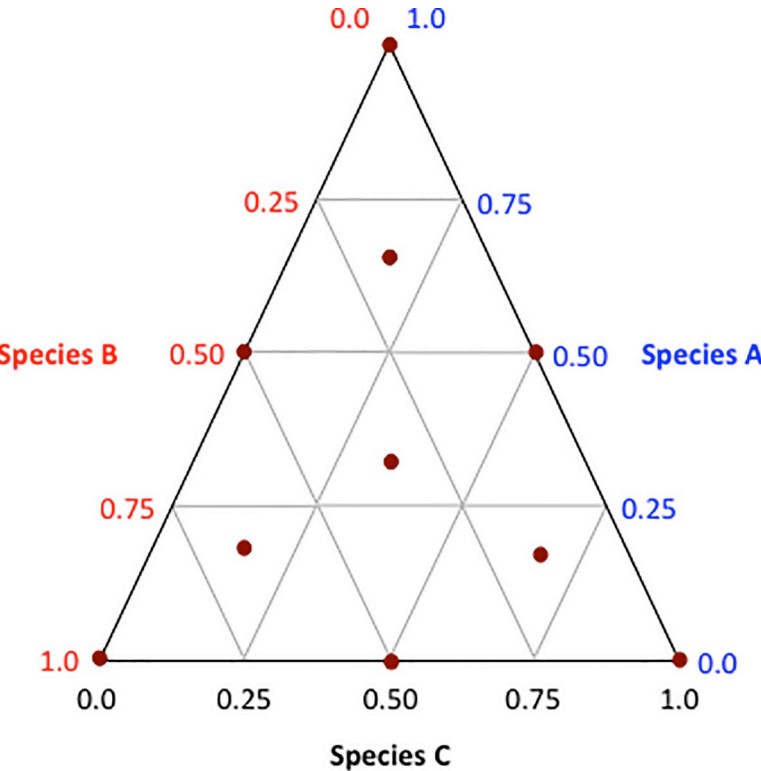

**Fig 1. Design space for a mixture of 3 conservation plant species.** The coordinates of points within the mixture space are expressed as proportions summing to 1. Each side of the triangular space corresponds to all of the proportions of a single plant species within the mixture. The red dots signify the proportional sampling points (treatments) measured in each experiment. Replications were performed over time.

rearing diets, Lapointe et al. [75] for assessing pheromone components, and to Lapointe et al. [76] in the identification of olfactory and gustatory stimulants of the Asian citrus psyllid.

We designed a 3-component mixture comprised of three conservation plant species. The design space was a triangle where each plant species corresponded to one of the vertices of the triangle; this design space represented the universe of all possible proportional combinations of these three species. Each species ranged from 0 percent to 100% of the mixture (**Fig 1**). It is important to keep in mind that the mixture components are not independent. For example, when the proportion of plant species 'A' was varied, the proportions of plant species 'B' and 'C' simultaneously varied to maintain the unity of the mixture; i.e., A + B + C = 1. This 'mixture design space', which was dictated by the proportionality of each of the three plant species, permitted us to track natural enemy occurrence over the surface of the design space. In our experiments, the plants were grown in planter boxes and each treatment contained the same number of planter boxes and same volume of plants within each planter box. This enabled us to keep the total number (amount) of plants constant while only varying proportion; therefore, the effects of amount and proportion were not confounded.

## Methods and materials

### Study site

Because this was a proof-of-concept study, we wanted to conduct the tests at an appropriate and manageable scale. We chose Heathcote Botanical Gardens, a 2.8-hectare public display

garden located in Fort Pierce, FL as the study site. Formerly the site of a commercial ornamental nursery, the vegetation of Heathcote Botanical Gardens is representative of residential areas and the urban/farmland interface in the Indian River citrus growing region. The site was secure, could provide water, and, importantly, was not treated with insecticides, which would have confounded the results. The 13 m x 60m study site was located in an open mown area periodically used as a parking lot and exposed to full sun throughout the day. It was surrounded by mature trees, primarily southern live oak (*Quercus virginiana* Mill.). The open conditions at the study site were representative of residential areas and citrus grove borders.

Prior to the start of the study, a ground cloth was placed on the 60 m x 13 m study site and covered with wood chip mulch to a depth of ca. 5 cm to provide a uniform substrate and discourage weed growth. Three experiments were conducted: late Spring-early Summer 2017 (June 2–27), Summer 2017 (August 1–25), and Summer 2018 (July 16-August 10). Permission to establish the study site was granted by the Executive Director of Heathcote Botanicals Gardens in 2017 and a cooperative agreement between Heathcote Botanical Gardens and USDA-ARS was authorized at that time.

## Plants

The following plant species were included in the experiments: Spring 2017: *Euphorbia milii* Des Moul. (crown-of-thorns), *Euphorbia heterophylla* L. (wild poinsettia), and *Phaseolus lunatus* L. (lima bean); Summer 2017: *E. milii*, *Fagopyrum esculentum* Moench (buckwheat), and *Chamaecrista fasciculata* Michx. Greene (partridge pea); and, Summer 2018: *E. milii*, *F. esculentum*, and *Portulaca umbraticola* Kunth (ornamental portulaca) (**Table 1**). Each of these plant species met the criteria for the 'right kind of plants' for biological control agents as described by Wäckers and van Rijn [32] in that they had either extrafloral nectaries or open shallow flowers whose nectaries were accessible to natural enemies such as *T. radiata* which possess short mouthparts and a limited floral foraging repertoire [77].

The study site was located in south Florida, an area designated in Plant Hardiness Zone 10b (average annual low temperature: 1.7˚- 4.4˚C). This zone is regarded as primarily subtropical. Therefore, the ability to grow well in this climate was an important criterion for selecting test plant species. All of the plant species tested in this study grow well and readily flower in south Florida and are easily cultivated there during the warm season. Two of the plants, *C. fasciculata* and *E. heterophylla* are native to Florida and have been shown to harbor natural enemies of *D. citri* in residential landscapes [77,78]. *E. milii* and *P. umbraticola* are common landscaping plants in south Florida and *T. radiata* fed on *E. milii* nectaries in a laboratory test [77].

**Table 1. Name, planting density, and characteristics of plant species used in the plant mixture arrays.**

| Scientific name | Common name | Family | No. plants/ planter box | Source | Characteristics |
| --- | --- | --- | --- | --- | --- |
| *Euphorbia milii* Des Moul. | Crown-of-thorns | Euphorbiacea | All tests: 3 plants | Local nursery | Ornamental perennial, blooms all year. Heat and drought tolerant. |
| *Portulaca umbraticola* Kunth | Ornamental portulaca | Portulacaceae | Summer 2018: 2 large or 3 small plants | Local nursery | Long-blooming ornamental annual. Heat and drought tolerant. |
| *Euphorbia heterophylla* L | Wild poinsettia | Euphorbiacea | Spring 2017: 6 plants | Grown from seed collected locally | Long-blooming annual. Native plant that can be weedy. |
| *Phaseolus lunatus* L | Lima bean | Fabaceae | Spring 2017: 5 plants | Johnny's Seeds, Albion ME | Annual garden legume with extra floral nectaries. Bush variety used. |
| *Fagopyrum esculentum* Moench | Buckwheat | Polygonaceae | All tests: 8 plants | Johnny's Seeds, Albion, ME | Quick-blooming annual. Green manure. |
| *Chamaecrista fasciculata* Michx. Greene | Partridge pea | Fabaceae | Summer 2017: 3 plants | Hancock Seed Company, Dade City, FL | Annual legume with extra floral nectaries. Native. |

Flowering buckwheat (*F. esculentum*) grows well in south Florida and has been frequently cited as being a floral host plant for biological control agents [32]. Lima bean (*P. lunatus*) is a garden vegetable that grows well in south Florida. All of the plant species are annuals with the exception of *E. milii*, which is a perennial. Drought tolerant species included *E. milii*, *P. umbraticola*, and *C. fasciculata*, which, along with *F. esculentum*, were used in the experiments during the summer.

## Planter boxes

Plants were grown in planter boxes made from 68 L gray plastic tote boxes (60.9 cm L x 41.9 cm W x 39.4 cm D) (Roughneck Box, Rubbermaid, Atlanta, GA, USA). Eight drainage holes were drilled in the bottom of each tote box. The boxes were filled to ca. 80% capacity with commercial potting soil (Farfard 4P Mix, SunGro Horticulture, Agawam, MA, USA). Plants were either grown from seed in 7.6 x 7.6 cm plastic containers or purchased from local nurseries (**Table 1**). Prior to sowing, seeds of *P. lunatus* and *C. fasciculata* were lightly scarified in a mortar and pestle partially filled with a sand slurry and inoculated with a *Rhizobium* culture (Verdesian Life Sciences, Cary, NC, USA). After transplanting into the planter boxes, 30 ml of slow-release fertilizer (Suncote 16-9-12, Everris International, Geldermalsen, The Netherlands) was spread on the surface of the soil to maintain soil fertility during the course of the experiments. The soil surface was covered with pine flake mulch to a depth of ca. 5 cm to help maintain soil moisture and prevent weeds. Plants were watered as needed, typically once or twice per week depending on the frequency of summer thunderstorms. Only a single species was grown in each planter box. Planting density was determined on the basis of expected size at maturity in an effort to provide a similar plant mass among planter boxes. The experiments were conducted once the plants flowered in the planter boxes.

## Test arrays

A mixture design was used to determine the effect of proportionality of three conservation plant species on natural enemies of the *D. citri*. Sample points in the design space were selected by D-optimality for quadratic modeling using Design-Expert® 10 (Stat-Ease, Inc.). The design included 6 model, 4 lack-of-fit, and 4 replicate points for a total of 14 mixture groups. Each mixture group was comprised of six planter boxes, having a proportion of 0.00, 0.17, 0.33, 0.66, or 1.00 of each plant species (**Fig 2, Table 2**). The number and type of proportional mixture groups included in the design was sufficient to construct a response surface model (**Fig 2**). The replicate mixtures provided sufficient degrees of freedom for estimating error across the design space, which was used in the ANOVA. Mixture treatments were randomly assigned within the study site. A control treatment, two groups with planters without plants, was also included in the study as a reference. A curtain made of loosely-woven burlap, extending upwards 1M from the top of the planter boxes, and supported by bamboo poles, was placed around the perimeters of the two control groups to approximate the visibility of the yellow sticky card traps in the interior of the planted groups. Mixture treatment groups were placed ca. 3 m apart from each other.

## Natural enemy sampling

The primary response variable was the frequency of *D. citri* natural enemies that occurred in each mixture group. To sample natural enemies within the test arrays, two 5.5 x 8.0 cm yellow sticky card traps (Alpha Scents, West Linn, OR, USA) were mounted on bamboo stakes at a height of 100 cm above the ground. The traps were placed 125 cm apart along the centerline of each mixture group. The cards were replaced twice each week; one sampling period lasted 4

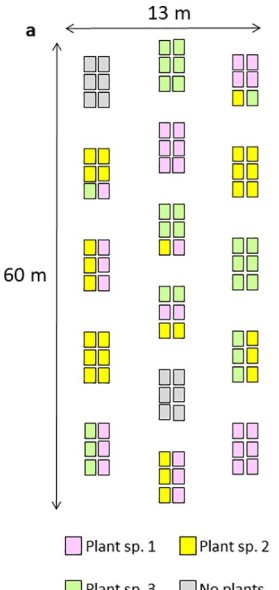
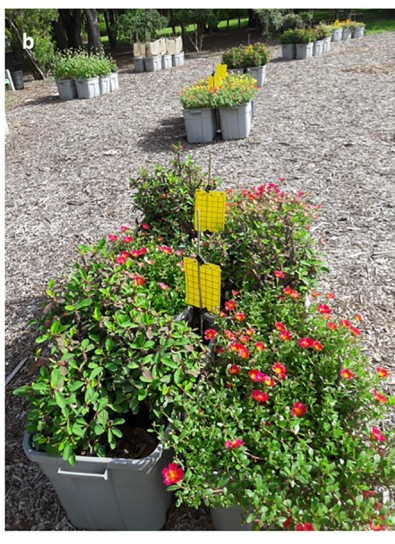

**Fig 2. a)** Diagrammatic representation of test array showing arrangement of planter boxes in each test mixture group. Each planter box contained only a single plant species and the number of planter boxes between mixture groups was kept constant. Colors indicate placement of each plant species within each mixture group to generate proportionalities. Arrows indicate dimensions of the study site. Not to scale. **b)** Photograph of the Summer 2018 Experiment showing a section of the study site and mixture treatment groups. The treatment group in the foreground is the 1:1 mixture of *Portulaca umbraticola* and *Euphorbia milii*.

days and the other 3 days. After the exposure period, the traps were covered with a transparent plastic cover and returned to the lab where they were stored at -18°C. The cards were examined with a dissection microscope for the presence of arthropod taxa known to attack *D. citri* [55–57]. Each trap sampling period represented a single replication of the experiment, and a total of seven replications were performed for each experiment.

**Table 2. Proportions of plant species in each treatment within the mixture design.** Note that the 'control' treatment (0,0,0) is not included here because there is no null mixture in the design space; it was included in the experiments only as a reference.

| Design point (treatment) | Point type | Plant species 1 | Plant species 2 | Plant species 3 |
|---|---|---|---|---|
| 1 | Lack of Fit | 0.66 | 0.17 | 0.17 |
| 2 | Model | 0 | 1 | 0 |
| 3 | Model | 0 | 0 | 1 |
| 4 | Model | 0 | 0.5 | 0.5 |
| 5 | Replicate | 1 | 0 | 0 |
| 6 | Lack of Fit | 0.17 | 0.66 | 0.17 |
| 7 | Lack of Fit | 0.17 | 0.17 | 0.66 |
| 8 | Lack of Fit | 0.33 | 0.33 | 0.33 |
| 9 | Model | 0.5 | 0.5 | 0 |
| 10 | Replicate | 0 | 1 | 0 |
| 11 | Model | 1 | 0 | 0 |
| 12 | Replicate | 0.5 | 0.5 | 0 |
| 13 | Replicate | 0 | 0 | 1 |
| 14 | Model | 0.5 | 0 | 0.5 |

## Analysis

The data from each collection period performed during a single experiment (Spring 2017, Summer 2017, Summer 2018) were pooled for analysis. A 3-component mixture design was constructed to identify plant mixtures that optimized the frequency of occurrence of natural enemies of *D. citri*. Each measured response was analyzed using Scheffé polynomial regression followed by ANOVA. Models that resulted in the best clustering of the three $R^2$ values ($R^2$, adjusted $R^2$, and predicted $R^2$) and minimum number of significant terms were selected. The software application Design-Expert® 10 (Stat-Ease, Inc.) was used for experimental design construction, data and model diagnostics and evaluation, ANOVA, and graphics. An analysis was performed on each individual predator group (coccinellids, predatory hemipterans, and *Blattella asahinai* Mizukubo) as well as a group comprised of all predators combined.

## Results

### Occurrence of natural enemies

Over the course of the study, three taxonomic groups of natural enemies of *D. citri* were collected from the plant mixture arrays: 1) Several species of coccinellids; 2) Three families of predatory hemipterans; and, 3) *B. asahinai*, an invasive roach native to Japan [79], and which has been implicated as a predator of *D. citri* [56] (**Fig 3**). In total, 11 large- and medium-sized coccinellid species were collected from the plant mixture arrays (**Table 3**). *Cryptolaemus montrouzieri* Mulsant and other small-bodied species from the genera *Diomus* and *Scymnus* were grouped together. Across all three experiments, the most numerous coccinellid species collected were *Coleophora inaequalis* Fabricius, (variable ladybeetle) (150 specimens), *Harmonia axyridis* Pallis, (multicolored Asian ladybeetle) (142 specimens), and *Cycloneda sanguinea* Linnaeus (spotless ladybeetle) (96 specimens). A total of 308 small-bodied species were collected. Of the predatory hemipterans, anthocorids were the most abundant (275 specimens) followed by nabids (92 specimens) (**Table 4**). The exotic roach, *B. asahinai*, was very abundant; a total of 1275 specimens were collected across all three experiments (**Table 5**).

There were distinct differences in predator abundance across the experiments. A total of 1580 predator specimens collected during the Spring 2017 experiment, 586 specimens collected during the Summer 2017 experiment, and 278 specimens collected during the Summer 2018 experiment (**Fig 3**). Hoverflies (Syrphidae) and lacewings (Chrysopidae), which are prominent predators of *D. citri* [55–57], and the parasitoid *T. radiata* were absent in the collections.

### Plant mixture effects on natural enemy occurrence

Response surface analysis revealed some proportionality effects (nonlinear blending) of individual plant species on predator occurrence. This effect was observed in the combine predators group in Spring 2017. In the Summer 2017 and 2018 experiments, no nonlinear blending effects observed, only the effect of the linear mixture, which is an amount effect. The linear mixture is the response at the vertices (i.e., a comparison of the 100% points, also referred to as 'pure blends').

In the Spring 2017 Experiment, the occurrence of all three taxonomic groups, as well as that of the combined predators group, were influenced by the proportionality of plant species within the test mixtures (**Table 6**, **Fig 4**). In each case, the ANOVAs showed that the overall models were highly significant ($P < 0.008$) indicating significant component effects (**Table 6**). *P. lunatus* (lima bean), which became heavily infested with cowpea aphid (*Aphis craccivora* C. L. Koch), was a primary driver of predator occurrence in this experiment (ANOVA: coccinellids: $P = 0.0799$; predatory hemipterans: $P = 0.0373$; combined predators: $P = 0.0175$). This was

**Table 3. Mean numbers of coccinellid species collected from plant mixture treatments per sampling period during each experiment.** The total number of each coccinellid species collected during each experiment are shown in right hand column while the total numbers of coccinellids collected from each plant mixture treatment are shown in the bottom rows.

**SPRING 2017**

| Coccinellid species | No plant control (2 arrays) | 100% Phaseolus (2 arrays) | 100% E. milli (2 arrays) | 100% E. heterophylla (2 arrays) | 33% Each (1 array) | 50% Phaseolus 50% E. milii (2 arrays) | 50% Phaseolus 50% E. heterophylla (1 array) | 50% E. milii 50% E. heterophylla (1 array) | 67% Phaseolus (1 array) | 67% E. milii (1 array) | 67% E. heterophylla (1 array) | Total number collected in experiment |
|---|---|---|---|---|---|---|---|---|---|---|---|---|
| *Coleophora inaequalis* | 0.1 | 3.3 | 0.1 | 0.3 | 0.8 | 1.6 | 1.4 | 0.0 | 2.8 | 0.5 | 1.0 | **135** |
| Small-bodied species | 0.9 | 1.8 | 0.6 | 1.1 | 0.6 | 1.0 | 1.8 | 1.3 | 3.9 | 0.8 | 0.9 | **158** |
| *Harmonia axyridis* | 0.4 | 0.9 | 0.6 | 0.6 | 1.3 | 2.1 | 0.1 | 0.0 | 2.9 | 1.1 | 0.9 | **124** |
| *Cycloneda sanguinea* | 0.2 | 0.4 | 0.3 | 0.1 | 0.1 | 1.0 | 0.6 | 0.6 | 3.0 | 1.0 | 0.5 | **77** |
| *Azya orbigera* | 0.1 | 0.8 | 0.2 | 0.1 | 0.8 | 0.5 | 0.0 | 0.3 | 0.6 | 0.4 | 0.6 | **46** |
| *Olla v-nigrum* | 0.0 | 0.1 | 0.0 | 0.1 | 0.0 | 0.0 | 0.0 | 0.0 | 0.0 | 0.0 | 0.0 | **3** |
| *Chilocorus spp.* | 0.0 | 0.1 | 0.1 | 0.0 | 0.0 | 0.0 | 0.1 | 0.1 | 0.0 | 0.0 | 0.1 | **6** |
| *Curinus coerulens* | 0.0 | 0.0 | 0.1 | 0.0 | 0.0 | 0.0 | 0.0 | 0.0 | 0.1 | 0.0 | 0.0 | **2** |
| *Exochomus childreni* | 0.0 | 0.0 | 0.0 | 0.1 | 0.0 | 0.0 | 0.0 | 0.0 | 0.0 | 0.0 | 0.0 | **1** |
| *Brachiacantha dentipes* | 0.0 | 0.0 | 0.1 | 0.0 | 0.0 | 0.1 | 0.0 | 0.0 | 0.0 | 0.0 | 0.1 | **4** |
| **Total no. collected** | **27** | **116** | **33** | **35** | **28** | **98** | **32** | **18** | **106** | **30** | **33** | **556** |

**SUMMER 2017**

| Coccinellid species | No plant control (2 arrays) | 100% Fagopyrum (2 arrays) | 100% E. milli (2 arrays) | 100% Chamaecrista (2 arrays) | 33% Each (1 array) | 50% Fagopyrum 50% E. milii (2 arrays) | 50% Fagopyrum 50% Chamaecrista (1 array) | 50% E. milii 50% Chamaecrista (1 array) | 67% Fagopyrum (1 array) | 67% E. milii (1 array) | 67% Chamaecrista (1 array) | Total number collected in experiment |
|---|---|---|---|---|---|---|---|---|---|---|---|---|
| *Coleophora inaequalis* | 0.0 | 0.0 | 0.1 | 0.2 | 0.0 | 0.0 | 0.0 | 0.0 | 0.0 | 0.0 | 0.0 | **4** |
| Small-bodied species | 0.6 | 1.1 | 0.9 | 1.4 | 0.9 | 0.8 | 1.0 | 1.1 | 0.6 | 0.3 | 1.4 | **103** |
| *Harmonia axyridis* | 0.0 | 0.0 | 0.2 | 0.0 | 0.0 | 0.0 | 0.0 | 0.0 | 0.0 | 0.4 | 0.1 | **7** |
| *Cycloneda sanguinea* | 0.0 | 0.0 | 0.2 | 0.3 | 0.1 | 0.1 | 0.0 | 0.0 | 0.0 | 0.4 | 0.1 | **14** |
| *Azya orbigera* | 0.0 | 0.0 | 0.1 | 0.1 | 0.1 | 0.0 | 0.1 | 0.0 | 0.4 | 0.3 | 0.0 | **10** |
| *Olla v-nigrum* | 0.0 | 0.0 | 0.0 | 0.0 | 0.0 | 0.0 | 0.0 | 0.0 | 0.0 | 0.0 | 0.0 | **0** |
| *Chilocorus spp.* | 0.0 | 0.0 | 0.0 | 0.0 | 0.0 | 0.0 | 0.0 | 0.0 | 0.0 | 0.0 | 0.0 | **0** |
| *Curinus coerulens* | 0.0 | 0.0 | 0.0 | 0.0 | 0.0 | 0.0 | 0.0 | 0.0 | 0.0 | 0.0 | 0.0 | **0** |
| *Exochomus childreni* | 0.0 | 0.1 | 0.0 | 0.0 | 0.0 | 0.0 | 0.0 | 0.1 | 0.0 | 0.0 | 0.0 | **2** |

*(Continued)*

Table 3. (Continued)

|  |  |  |  |  |  |  |  |  |  |  |  |  |
|---|---|---|---|---|---|---|---|---|---|---|---|---|
| *Brachiacantha dentipes* | 0.0 | 0.3 | 0.2 | 0.1 | 0.0 | 0.2 | 0.1 | 0.1 | 0.1 | 0.1 | 0.1 | 0.0 | 16 |
| *B. decora* | 0.0 | 0.0 | 0.0 | 0.0 | 0.0 | 0.0 | 0.1 | 0.0 | 0.0 | 0.0 | 0.0 | 0.0 | 1 |
| *Hyperaspis bigeminata* | 0.0 | 0.0 | 0.0 | 0.0 | 0.0 | 0.1 | 0.0 | 0.0 | 0.0 | 0.0 | 0.0 | 0.0 | 1 |
| **Total no. collected** | **9** | **20** | **24** | **30** | **8** | **17** | **9** | **10** | **10** | **8** | **11** | **12** | **158** |

**SUMMER 2018**

| | Plant mixture treatment | | | | | | | | | | | Total number collected in experiment |
|---|---|---|---|---|---|---|---|---|---|---|---|---|
| | No plant control (2 arrays) | 100% *Fagopyrum* (2 arrays) | 100% *E. milli* (2 arrays) | 100% *Portulaca* (2 arrays) | 33% Each (1 array) | 50% *Fagopyrum* 50% *E. milii* (2 arrays) | 50% *Fagopyrum* 50% *Portulaca* (1 array) | 50% *E. milii* 50% *Portulaca* 1 array) | 67% *Fagopyrum* (1 array) | 67% *E. milii* (1 array) | 67% *Portulaca* (1 array) | |
| *Coleophora inaequalis* | 0.0 | 0.1 | 0.2 | 0.2 | 0.1 | 0.0 | 0.0 | 0.0 | 0.0 | 0.3 | 0.1 | 11 |
| *Small-bodied species* | 0.2 | 0.1 | 0.3 | 0.3 | 0.1 | 0.5 | 0.4 | 0.5 | 1.0 | 1.0 | 0.3 | 47 |
| *Harmonia axyridis* | 0.0 | 0.0 | 0.0 | 0.3 | 0.0 | 0.0 | 0.0 | 0.1 | 0.1 | 0.0 | 0.5 | 11 |
| *Cycloneda sanguinea* | 0.1 | 0.0 | 0.0 | 0.1 | 0.1 | 0.0 | 0.0 | 0.0 | 0.1 | 0.0 | 0.0 | 5 |
| *Azya orbigera* | 0.0 | 0.0 | 0.0 | 0.0 | 0.0 | 0.0 | 0.0 | 0.0 | 0.0 | 0.0 | 0.0 | 0 |
| *Olla v-nigrum* | 0.0 | 0.0 | 0.0 | 0.0 | 0.0 | 0.0 | 0.0 | 0.0 | 0.0 | 0.0 | 0.0 | 0 |
| *Chilocorus spp.* | 0.0 | 0.0 | 0.0 | 0.0 | 0.0 | 0.0 | 0.0 | 0.0 | 0.0 | 0.0 | 0.0 | 0 |
| *Curinus coerulens* | 0.0 | 0.0 | 0.0 | 0.0 | 0.0 | 0.0 | 0.0 | 0.0 | 0.0 | 0.0 | 0.0 | 0 |
| *Exochomus childreni* | 0.0 | 0.0 | 0.0 | 0.0 | 0.0 | 0.0 | 0.0 | 0.0 | 0.0 | 0.0 | 0.0 | 0 |
| *Brachiacantha dentipes* | 0.0 | 0.1 | 0.0 | 0.0 | 0.0 | 0.0 | 0.0 | 0.0 | 0.0 | 0.0 | 0.1 | 2 |
| **Total no. collected** | **4** | **3** | **7** | **15** | **3** | **8** | **3** | **5** | **10** | **10** | **8** | **76** |

**Table 4. Mean numbers of predatory hemipterans collected from plant mixture treatments per sampling period during each experiment.** Total numbers of specimens from each family are shown in right hand column. The total numbers of all predatory hemipterans collected in each plant mixture treatment are shown in bottom rows.

### SPRING 2017

| Family | No plant control (2 arrays) | 100% Phaseolus (2 arrays) | 100% E. milii (2 arrays) | 100% E. heterophylla (2 arrays) | 33% Each (1 array) | 50% Phaseolus 50% E. milii (2 arrays) | 50% Phaseolus 50% E. heterophylla (1 array) | 50% E. milii 50% E. heterophylla (1 array) | 67% Phaseolus (1 array) | 67% E. milii (1 array) | 67% E. heterophylla (1 array) | Total number collected in experiment |
|---|---|---|---|---|---|---|---|---|---|---|---|---|
| Anthocoridae | 0.6 | 0.8 | 0.6 | 0.3 | 0.1 | 0.4 | 0.6 | 1.1 | 0.8 | 0.4 | 0.1 | **75** |
| Nabidae | 0.5 | 0.1 | 1.6 | 0.4 | 0.6 | 0.3 | 0.4 | 0.4 | 0.3 | 1.3 | 0.8 | **68** |
| Reduvidae | 0.0 | 0.1 | 0.1 | 0.1 | 0.6 | 0.1 | 0.0 | 0.1 | 0.1 | 0.1 | 0.1 | **15** |
| **Total no. collected** | **17** | **15** | **37** | **13** | **11** | **13** | **8** | **13** | **9** | **14** | **8** | **158** |

### SUMMER 2017

| Family | No plant control (2 arrays) | 100% Fagopyrum (2 arrays) | 100% E. milii (2 arrays) | 100% Chamaecrista (2 arrays) | 33% Each (1 array) | 50% Fagopyrum 50% E. milii (2 arrays) | 50% Fagopyrum 50% Chamaecrista (1 array) | 50% E. milii 50% Chamaecrista (1 array) | 67% Fagopyrum (1 array) | 67% E. milii (1 array) | 67% Chamaecrista (1 array) | Total number collected in experiment |
|---|---|---|---|---|---|---|---|---|---|---|---|---|
| Anthocoridae | 1.4 | 2.3 | 1.8 | 1.6 | 1.6 | 1.5 | 2.1 | 1.1 | 0.6 | 1.4 | 0.4 | **171** |
| Nabidae | 0.1 | 0.4 | 0.1 | 0.1 | 0.0 | 0.0 | 0.0 | 0.0 | 0.0 | 0.0 | 0.3 | **11** |
| Reduvidae | 0.1 | 0.4 | 0.1 | 0.1 | 0.0 | 0.0 | 0.0 | 0.0 | 0.0 | 0.0 | 0.1 | **10** |
| **Total no. collected** | **21** | **42** | **29** | **25** | **11** | **21** | **15** | **8** | **4** | **10** | **6** | **192** |

### SUMMER 2018

| Family | No plant control (2 arrays) | 100% Fagopyrum (2 arrays) | 100% E. milii (2 arrays) | 100% Portulaca (2 arrays) | 33% Each (1 array) | 50% Fagopyrum 50% E. milii (2 arrays) | 50% Fagopyrum 50% Portulaca (1 array) | 50% E. milii 50% Portulaca (1 array) | 67% Fagopyrum (1 array) | 67% E. milii (1 array) | 67% Portulaca (1 array) | Total number collected in experiment |
|---|---|---|---|---|---|---|---|---|---|---|---|---|
| Anthocoridae | 0.1 | 0.1 | 0.0 | 0.1 | 0.0 | 0.1 | 0.1 | 0.1 | 0.1 | 0.1 | 0.0 | **11** |
| Nabidae | 0.1 | 0.1 | 0.3 | 0.1 | 0.1 | 0.0 | 0.0 | 0.0 | 0.0 | 0.3 | 0.0 | **13** |
| Reduvidae | 0.0 | 0.1 | 0.0 | 0.2 | 0.0 | 0.1 | 0.0 | 0.0 | 0.0 | 0.0 | 0.0 | **5** |
| **Total no. collected** | **4** | **4** | **5** | **7** | **1** | **2** | **1** | **1** | **1** | **3** | **0** | **29** |

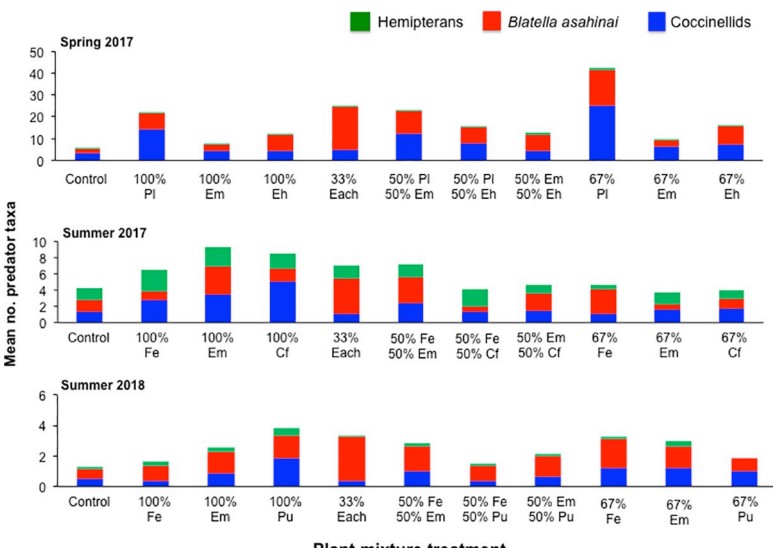

**Fig 3. Mean number of natural enemies from different taxa occurring in each conservation plant mixture group during each experiment.** Key to plant species abbreviations: Spring 2017: Pl = *Phaseolus lunatus*, Em = *Euphorbia milii*, Eh = *E. heterophylla*; Summer 2017: Fe = *Fagopyrum esculentum*, Cf = *Chamaecrista fasciculata*, Em = *E. milii*; Summer 2018: Pu = *Portulaca umbraticola*, Fe = *Fagopyrum esculentum*, Em = *E. milii*.

**Table 5. Total numbers of *Blattella asahinai* collected from plant mixture treatments during each experiment.** Total numbers of specimens collected is shown in right hand column.

| SPRING 2017 | Plant mixture treatment | | | | | | | | | | | |
|---|---|---|---|---|---|---|---|---|---|---|---|---|
| | No plant control (2 arrays) | 100% *Phaseolus* (2 arrays) | 100% *E. milii* (2 arrays) | 100% *E. heterophylla* (2 arrays) | 33% Each (1 array) | 50% *Phaseolus* 50% *E. milii* (2 arrays) | 50% *Phaseolus* 50% *E. heterophylla* (1 array) | 50% *E. milii* 50% *E. heterophylla* (1 array) | 67% *Phaseolus* (1 array) | 67% *E. milii* (1 array) | 67% *E. heterophylla* (1 array) | Total number collected in experiment |
| **Total no. collected** | 26 | 108 | 38 | 102 | 148 | 145 | 56 | 52 | 115 | 19 | 57 | 866 |
| SUMMER 2017 | Plant mixture treatment | | | | | | | | | | | |
| | No plant control (2 arrays) | 100% *Fagopyrum* (2 arrays) | 100% *E. milii* (2 arrays) | 100% *Chamaecrista* (2 arrays) | 33% Each (1 array) | 50% *Fagopyrum* 50% *E. milii* (2 arrays) | 50% *Fagopyrum* 50% *Chamaecrista* (1 array) | 50% *E. milii* 50% *Chamaecrista* (1 array) | 67% *Fagopyrum* (1 array) | 67% *E. milii* (1 array) | 67% *Chamaecrista* (1 array) | Total number collected in experiment |
| **Total no. collected** | 21 | 14 | 48 | 24 | 30 | 44 | 5 | 15 | 21 | 5 | 9 | 236 |
| SUMMER 2018 | Plant mixture treatment | | | | | | | | | | | |
| | No plant control (2 arrays) | 100% *Fagopyrum* (2 arrays) | 100% *E. milii* (2 arrays) | 100% *Portulaca* (2 arrays) | 33% Each (1 array) | 50% *Fagopyrum* 50% *E. milii* (2 arrays) | 50% *Fagopyrum* 50% *Portulaca* (1 array) | 50% *E. milii* 50% *Portulaca* (1 array) | 67% *Fagopyrum* (1 array) | 67% *E. milii* (1 array) | 67% *Portulaca* (1 array) | Total number collected in experiment |
| **Total no. collected** | 10 | 16 | 22 | 24 | 23 | 26 | 8 | 11 | 15 | 11 | 7 | 173 |

**Table 6. Reduced quadratic response surface model, including diagnostic statistics, for natural enemy occurrence in response to mixture arrays containing three plant species.** Note that the "linear model" term is used in a mixture ANOVA because the individual components are not independent and are considered together (compared to each other). The effect of a mixture component is defined by a gradient (or slope) in some specified direction, and these effects are shown in the trace plots (Figs 4 & 5). The terms AB, AC, and BC are quadratic blending or nonlinear blending terms in mixture models and show if the two components are exhibiting a synergism or antagonism–i.e., a response not predicted by the simple additive blending in the linear mixture model. They are not "interaction" terms because the components are not independent.

| Predator Group | Source | Spring 2017 (May-June)[a] | | | | | Summer 2017 (August)[a] | | | | | Summer 2018 (July-August)[a] | | | | |
|---|---|---|---|---|---|---|---|---|---|---|---|---|---|---|---|---|
| | | Sum of Squares | df | Mean Square | F Value | P value | Sum of Squares | df | Mean Square | F Value | P value | Sum of Squares | df | Mean Square | F Value | P value |
| **Coccinellids** | Model | 1 | 3 | 0 | 12 | **0.0012** | 1 | 2 | 0 | 2 | 0.1417 | 1 | 2 | 1 | 6 | **0.0201** |
| | Linear Mixture | 1 | 2 | 1 | 16 | **0.0008** | 1 | 2 | 0 | 2 | 0.1417 | 1 | 2 | 1 | 6 | **0.0201** |
| | AB (*Phaseolus* x *E. milii*) | 0 | 1 | 0 | 4 | **0.0799** | | | | | | | | | | |
| | Residual | 0 | 10 | 0 | | | 2 | 11 | 0 | | | 1 | 11 | 0 | | |
| | Lack of Fit | 0 | 6 | 0 | 1 | 0.3962 | 1 | 7 | 0 | 0 | 0.9129 | 1 | 7 | 0 | 3 | 0.1970 |
| | Pure Error | 0 | 4 | 0 | | | 1 | 4 | 0 | | | 0 | 4 | 0 | | |
| | Cor Total | 2 | 13 | | | | 3 | 13 | | | | 2 | 13 | | | |
| | $R^2$ | 0.78 | | | | | 0.30 | | | | | 0.51 | | | | |
| | $R^2_{adj}$ | 0.71 | | | | | 0.17 | | | | | 0.42 | | | | |
| | $R^2_{pred}$ | 0.58 | | | | | -0.33 | | | | | 0.26 | | | | |
| ***Blattella asahinai*** | Model | 0 | 3 | 0 | 7 | **0.0076** | 5.9 | 2.0 | 3.0 | 1.5 | 0.2667 | 0.1 | 2.0 | 0.0 | 0.1 | 0.9251 |
| | Linear Mixture | 0 | 2 | 0 | 7 | **0.0115** | 5.9 | 2.0 | 3.0 | 1.5 | 0.2667 | 0.1 | 2.0 | 0.0 | 0.1 | 0.9251 |
| | AB (*Phaseolus* x *E. milii*) | 0 | 1 | 0 | 7 | **0.0250** | | | | | | | | | | |
| | Residual | 0 | 10 | 0 | | | 22 | 11 | 2 | | | 5 | 11 | 0 | | |
| | Lack of Fit | 0 | 6 | 0 | 2 | 0.2456 | 12 | 7 | 2 | 1 | 0.6926 | 3 | 7 | 0 | 1 | 0.4615 |
| | Pure Error | 0 | 4 | 0 | | | 10 | 4 | 2 | | | 2 | 4 | 0 | | |
| | Cor Total | 0 | 13 | | | | 28 | 13 | | | | 5 | 13 | | | |
| | $R^2$ | 0.68 | | | | | 0.21 | | | | | 0.01 | | | | |
| | $R^2_{adj}$ | 0.59 | | | | | 0.07 | | | | | -0.17 | | | | |
| | $R^2_{pred}$ | 0.43 | | | | | -0.36 | | | | | -0.51 | | | | |
| **Predatory Hemiptera[1]** | Model | 1 | 4 | 0 | 7 | **0.0067** | 2 | 3 | 1 | 1 | 0.3518 | 6 | 3 | 2 | 2 | 0.2310 |
| | Linear Mixture | 1 | 2 | 0 | 9 | **0.0064** | 1 | 2 | 0 | 0 | 0.6501 | 2 | 2 | 1 | 1 | 0.4926 |
| | AB (*Phaseolus* x *E. milii*) | 0 | 1 | 0 | 6 | **0.0373** | | | | | | | | | | |
| | BC (*E. milii* x *E. heterophylla*) | 0 | 1 | 0 | 3 | **0.0948** | | | | | | | | | | |
| | Residual | 0 | 9 | 0 | | | 6 | 10 | 1 | | | 12 | 10 | 1 | | |
| | Lack of Fit | 0 | 5 | 0 | 0 | 0.9109 | 3 | 6 | 0 | 1 | 0.7239 | 7 | 6 | 1 | 1 | 0.6374 |
| | Pure Error | 0 | 4 | 0 | | | 3 | 4 | 1 | | | 6 | 4 | 1 | | |
| | Cor Total | 2 | 13 | | | | 8 | 13 | | | | 18 | 13 | | | |
| | $R^2$ | 0.76 | | | | | 0.27 | | | | | 0.34 | | | | |
| | $R^2_{adj}$ | 0.66 | | | | | 0.05 | | | | | 0.14 | | | | |
| | $R^2_{pred}$ | 0.39 | | | | | -0.70 | | | | | -0.12 | | | | |
| **Combined Predators[2]** | Model | 0 | 3 | 0 | 10 | **0.0021** | 37 | 3 | 12 | 8 | **0.0057** | 0 | 2 | 0 | 7 | **0.0134** |
| | Linear Mixture | 0 | 2 | 0 | 11 | **0.0026** | 29 | 2 | 14 | 9 | **0.0059** | 0 | 2 | 0 | 7 | **0.0134** |
| | AB (*Phaseolus* x *E. milii*) | 0 | 1 | 0 | 8 | **0.0175** | | | | | | | | | | |
| | Residual | 0 | 10 | 0 | | | 16 | 10 | 2 | | | 0 | 11 | 0 | | |
| | Lack of Fit | 0 | 6 | 0 | 10 | 0.0202 | 15 | 6 | 2 | 8 | 0.0346 | 0 | 7 | 0 | 1 | 0.5892 |

*(Continued)*

**Table 6.** (Continued)

| Predator Group | Source | Spring 2017 (May-June)[a] | | | | | Summer 2017 (August)[a] | | | | | Summer 2018 (July-August)[a] | | | | |
|---|---|---|---|---|---|---|---|---|---|---|---|---|---|---|---|---|
| | | Sum of Squares | df | Mean Square | F Value | P value | Sum of Squares | df | Mean Square | F Value | P value | Sum of Squares | df | Mean Square | F Value | P value |
| | Pure Error | 0 | 4 | 0 | | | 1 | 4 | 0 | | | 0 | 4 | 0 | | |
| | Cor Total | 0 | 13 | | | | 53 | 13 | | | | 0 | 13 | | | |
| | $R^2$ | 0.76 | | | | | 0.70 | | | | | 0.54 | | | | |
| | $R^2_{adj}$ | 0.68 | | | | | 0.61 | | | | | 0.46 | | | | |
| | $R^2_{pred}$ | 0.63 | | | | | 0.50 | | | | | 0.31 | | | | |
| [1]Transformed Log10 (Total + 0.001) | | [a]Plant species used in mixtures: | | | | | [a]Plant species used in mixtures: | | | | | [a]Plant species used in mixtures: | | | | |
| [2]Transformed Log10(Total) | | *Phaseolus lunatus* | | | | | *Chamaecrista fasciculata* | | | | | *Portulaca umbraticola* | | | | |
| | | *Poinsettia heterophylla* | | | | | *Fagopyrum esculentum* | | | | | *Fagopyrum esculentum* | | | | |
| | | *Euphorbia milii* | | | | | *Euphorbia milii* | | | | | *Euphorbia milii* | | | | |

also shown by the consistently positive slope for this species in the trace plots of Cox-effects, which estimates the effects of increasing the proportion of one plant species in relation to a reference blend while keeping constant the relative proportions of the other two plant species (**Fig 4**). Trace plots are also called component effects plots. In our analyses, we used the 0.33 proportion of each plant species as the reference blend–this is the geometric center (1/3, 1/3, 1/3) of the triangle design space. The ANOVAs also revealed an effect of *E. milii* (crown-of-thorns) on the occurrence of *B. asahinai* (P = 0.0250), and the combined predators group. The positive slope for *E. heterophylla* (wild poinsettia) in the trace plots of Cox-effects, as well as the ANOVA (P = 0.0948), indicated a positive effect of this plant species on the occurrence of predatory hemipterans (**Fig 4D**).

In the Spring 2017 Experiment, the three $R^2$ statistics ($R^2$, $R^2_{adj}$ and $R^2_{pred}$) for the coccinellids and the combined predator group were clustered with a difference less than 0.2 (**Table 6**). Clustering of the $R^2$ statistics indicated that the amount of variation was within acceptable limits and did not interfere with the precision of the model. For the predatory hemipterans and *B. asahinai*, the $R^2$ statistics indicated that the models did not explain all of the observed variance. In the Summer 2017 Experiment, no nonlinear blending effects were observed for any of the predator groups (**Table 6**). However, a linear response was observed for *E. milii* indicating that increasing amounts of this plant drove total predator occurrence in the test mixtures (**Fig 5A & 5B**); *C. fasciculata* (partridge pea) and *F. esculentum* (flowering buckwheat) had little influence in this regard.

Likewise, in the Summer 2018 Experiment, a linear response indicated that *P. umbraticola* (ornamental portulaca) significantly influenced the occurrence of coccinellids and the combined predators group (**Table 6**, **Fig 5**). The $R^2$ statistics were not tightly clustered, indicating that the models did not explain all of the observed variance for the coccinellids and combined predators group. Interestingly, the trace plots of Cox effects showed a negative slope for *F. esculentum*, (buckwheat) indicating that this species had a negative effect on natural enemy occurrence in the test mixture plots. This is contrary to other studies demonstrating that these flowers support biocontrol insects [24, 80–82]. The buckwheat flowers were visited extensively by honeybees and scavenger flies, which likely originated from a large refuse container at a restaurant adjacent to the study site. Foraging by these two groups may have competitively limited the buckwheat's floral resources for natural enemies [33,82]. Contrary to the results obtained in the Summer 2017 experiment, *E. milii* showed no positive effect on predator occurrence in the Summer 2018 experiment.

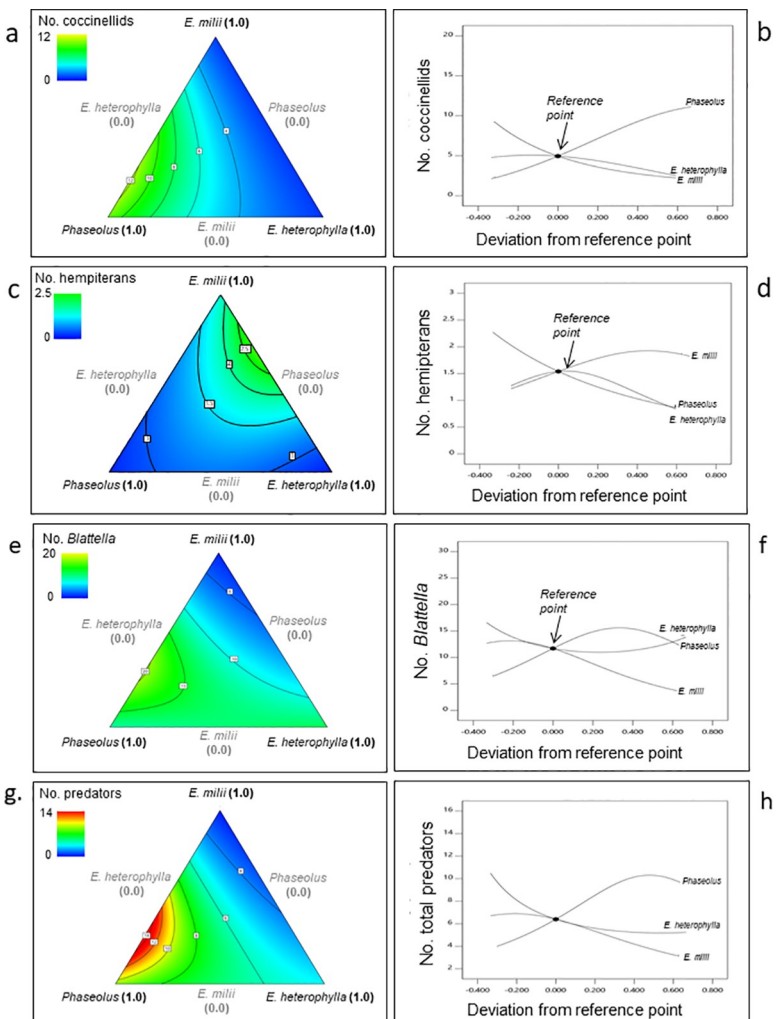

**Fig 4. Response surface plots showing the influence of proportionality of plant species on the occurrence of insect predators in the Spring 2017 experiment.** The 3-component mixture was composed of *Euphorbia milii* (crown-of-thorns), *E. heterophylla* (wild poinsettia), and *Phaseolus lunatus* (lima bean). **a**) Contour plot of the number of coccinellids. **b**) Trace plot of the effects of *E. milii*, *P. lunatus*, and *E. heterophylla* on the number of coccinellids. **c**) Contour plot of the number of predatory hemipterans. **d**) Trace plot of the effects of the *E. milii*, *P. lunatus*, and *E. heterophylla* on the number of predatory hemipterans. **e**) Contour plot of the number of *B. asahinai*. **f**) Trace plot of the effects of the *E. milii*, *P. lunatus*, and *E. heterophylla* on the number of *B. asahinai*. **g**) Contour plot of the number of all predators combined. **g**) Trace plot of the effects of the *E. milii*, *P. lunatus*, and *E. heterophylla* on the number of all predators combined.

## Discussion

RSM analysis showed that natural enemy occurrence was influenced by the presence of certain plant species within the test mixtures, either through linear mixture effects or nonlinear blending effects. Linear mixture effects indicated that predator occurrence was driven by the amount of a particular plant species within a test mixture while nonlinear blending effects indicated an effect of plant species proportionality; in other words, the plant mixture itself had emergent properties that contributed to predator occurrence. Because this nonlinear blending increased the number of predators, a desired response, it is considered a synergy. The mechanisms underlying these emergent properties have yet to be determined, but likely arose from the presence of particular nutritional resources or microhabitat properties. The results here

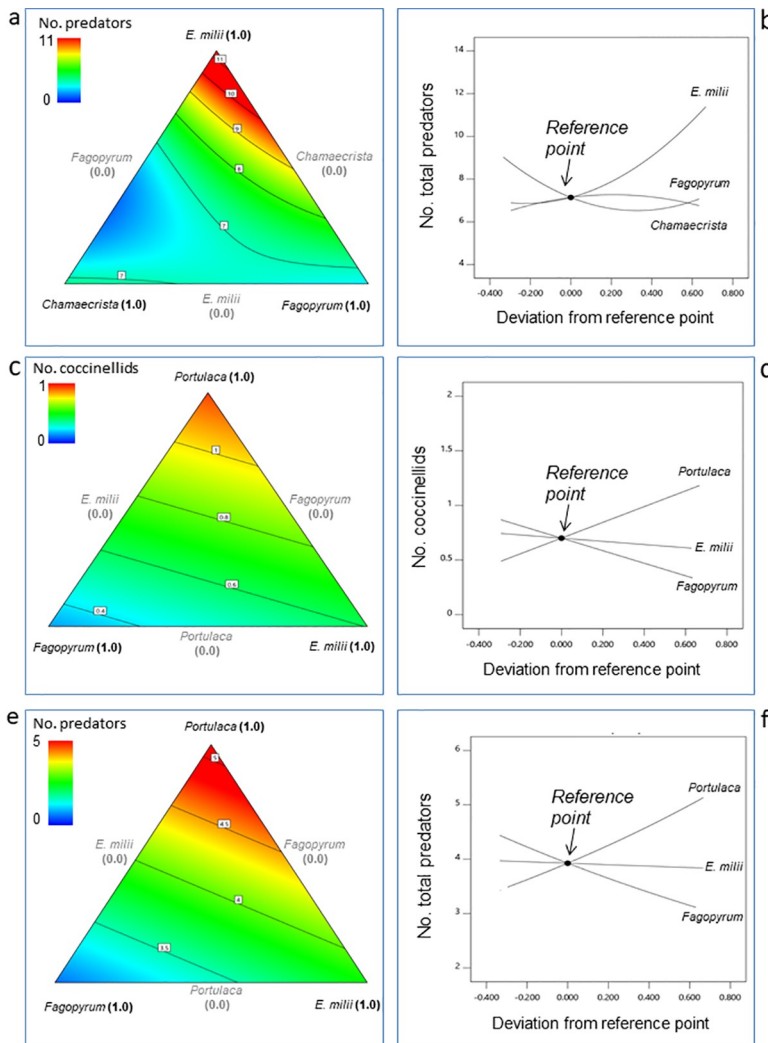

**Fig 5. Response surface plots of the occurrence of insect predators in the Summer 2017 and Summer 2018 experiments.** The 3-component mixture was composed of *Euphorbia milii* (crown-of-thorns), *Chamaecrista fasciculata* (partridge pea), and *Fagopyrum esculentum* (flowering buckwheat) in the Summer 2017 Experiment and *Euphorbia milii* (crown-of-thorns), *Portulaca umbraticola* (ornamental portulaca), and *Fagopyrum esculentum* (flowering buckwheat) in the Summer 2018 Experiment. **a**) Contour plot of the number of all predators combined in the Summer 2017 Experiment. **b**) Trace plot of the effects of the *E. milii*, *F. esculentum*, and *C. fasciculata* on the number of all predators combined. **c**) Contour plot of the number of coccinellids in the Summer 2018 Experiment. **d**) Trace plot of the effects of the *E. milii*, *F. esculentum*, and *P. umbraticola* on the number of coccinellids. **e**) Contour plot of the number of all predators combined. **f**) Trace plot of the effects of the *E. milii*, *F. esculentum*, and *P. umbraticola* on the number of all predators combined.

suggest that consideration should be given to the proportion of each conservation plant species when formulating a plant mixture to aid in conservation biological control.

Natural enemy abundance was highest in the Spring 2017 experiment and only in this experiment were nonlinear blending effects observed. *Phaseolus lunatus* (lima bean) was the primary driver of predator occurrence; this was likely due to a heavy infestation of aphids. Interestingly, predator occurrence was higher in test blocks having 66% v. 100% *P. lunatus* (**Table 6**, **Figs 3** & **4**). The nonlinear blending effect of *P. lunatus* and *E. milii* indicated that a mixture of these two species provided better habitat for predators than *P. lunatus* alone, and that together they formed a synergistic relationship. For predatory hemipterans, a strong

nonlinear blending synergy was observed for *E. milii* x *E. heterophylla* and the weaker effect observed for *E. milii* x *E. heterophylla* (**Table 6**, **Fig 5**). This indicated that a combination of all three plant species provided good habitat for these insects.

Predator occurrence decreased in subsequent experiments, which were conducted in the warmer summer months. In the Summer 2017 experiment, predator occurrence was only about a third of that observed in the Spring 2017 experiment and predator occurrence in the Summer 2018 experiment was less than half of that observed in the Summer 2017 experiment. In both summer experiments only linear mixture effects were observed, which indicated that predator occurrence was driven by the amount of *E. milii* (Summer 2017) or *P. umbraticola* (Summer 2018) in the test mixtures. We do not know why nonlinear blending effects were not observed in the two summer experiments. Possible explanations could be because of a low ambient levels of predators, due to the summer heat, or to some other factor(s) attributable to the plant species used in each mixture experiment. Another possibility is that, at low predator densities, larger plant mixture arrays are required to reveal nonlinear blending effects. Blaauw and Isaacs [83] found that natural enemy density, group richness, and diversity was greater in larger wildflower test plots than in smaller ones. A different experimental design, where the same plant mixtures are tested across seasons and with differently sized mixture arrays is needed to adequately explore this issue.

There were several unexpected outcomes of the study. The first was a banker plant effect, observed in the Spring 2017 experiment, in which an aphid infestation on *P. lunatus* was the primary driver of predator abundance. Our original concept was that plant-provided nutrients would drive predator occurrence in the test mixtures and *P. lunatus* was selected for the experiment because it has stipular extrafloral nectaries. The results here indicate that banker plant species that can support non-pestiferous alternate prey may be an important component of plant mixtures grown to support natural enemies of *D. citri*, as is the case in other systems [5,84].

The second unexpected outcome was the abundance of *B. asahinai* in the test plots. The preferred habitat of this exotic roach species is leaf litter in shaded areas [85]. Since the study site was covered with several cm of chipped wood mulch and was adjacent to a grove of oak trees, the appearance of this species should not have been surprising. However, it was very abundant, with a total of 1278 specimens collected over all experiments and comprising between 40 and 62% of all predator specimens collected. While the presence of mulch and the nearby oak grove may have been a requisite for the presence of this species, the results indicated that its occurrence in the mixture arrays was influenced by plant species composition. There was both a linear and nonlinear blending effect observed with *B. asahinai* with respect to arrays containing *P. lunatus*. Across experiments the numerically highest numbers of *B. asahinai* occurred in the mixture arrays comprised of equal proportions of each plant species (**Table 5**, **Fig 3**). Whether the roaches fed on the aphids or honeydew, or both, was not determined. Qureshi and Stansly [56] observed large numbers of *B. asahinai* caught in sticky barriers on citrus branches harboring *D. citri* and Pfannenstiel et al. [79] observed this species feeding heavily on sentinel moth eggs placed in soybean (*Glycine max* (L.) Merr.). However, while citrus trees in a commercial grove that were mulched had improved growth, psyllid densities in them were no different than unmulched trees [86]. Whether *B. asahinai* could significantly impact *D. citri* populations requires further examination.

The third unexpected outcome was the lack of syrphid flies and chrysopids collected during the study. The yellow sticky card traps may have been ineffectual in capturing these insects and future studies should include additional sampling methods such as visual inspections and malaise traps. Pan traps were not included as a sampling method because, in preliminary tests, frequent warm season thunderstorms caused them to overflow and precluded their use.

Alternatively, these insects may not have been abundant at the study site or during the time-frame when the experiments were conducted. A longer term study is needed to determine the importance of seasonal and annual variability on the abundance and diversity of syrphids and chrysopids at this study site.

Conservation biological control can be an effective approach for suppressing pests in citrus [68–69,87]. In the absence of frequent insecticidal applications, many types of predaceous arthropods attack *D. citri* in citrus [55,56,60]. More recently, Martini et al. [88] found significantly fewer *D. citri* on the edges of citrus groves with windbreaks as opposed to those without windbreaks and Tomaseto et al. [89] showed that trap crops planted near grove borders were effective for intercepting psyllids. Plants with nutritional resources assessable to the natural enemies of *D. citri* [77–78] could be incorporated alongside windbreaks or trap crops to add a conservation biological control component to an area-wide psyllid management program. To this end, whether natural enemies commute between such plantings and citrus trees will need to be determined [90].

Potentially, a variety of nectary and banker plant species could be used to support natural enemies of *D. citri*. Selection of conservation plants species will depend, in large part, on the target landscape where psyllid suppression is wanted. For example, aesthetic appeal will be an important attribute for conservation plants grown in residential landscapes while those grown in citrus groves may need to be drought tolerant and amenable to mechanized planting and cultivation. The purpose of the present study was to demonstrate that RSM could be used as a tool to evaluate the effectiveness of different plant mixtures with respect to the occurrence of the psyllid's predators. This information will permit us to select plant species or species mixtures that are optimal for promoting conservation biological control in residential landscapes, commercial groves, or other targeted landscapes.

Ultimately, we will need to determine if inclusion of conservation plants in the targeted landscape results in significant suppression of *D. citri* populations. In the future, we will evaluate plant mixtures that contain: 1) Apiaceous species, such as coriander (*Coriandrum sativum* L.) and dill (*Anethum graveolens* L.), species that are known to support biocontrol insects [21,91] but were not included in the present study; and, 2) Banker plants, which, as was shown by lima bean in this study, are likely to be an important component for a conservation biological control strategy in subtropical Florida. These studies will include additional measures, such as a timed observation component and gut content analysis, to determine the level of feeding by each predator group on the different nutritional resources (pollen, nectar, alternate prey) present within groups of test plants. We will also determine the importance of plant group size and planting arrangements in attracting and supporting natural enemies, especially during periods of low *D. citri* abundance and at different seasons of the year. Studies will evaluate whether a single strip of conservation plants placed along the grove border is as effective in suppressing psyllids as other types of planting arrangement, such as multiple strips planted at intervals throughout the grove or a cover crop grown across the grove understory. Finally, analyses, such as that conducted by Monzo and Stansly [68] which examined costs and benefits of conservation biological control across different yield-loss scenarios, will be needed to demonstrate the economic feasibility of implementing conservation plantings as a means of suppressing *D. citri*.

## Supporting information

**S1 Table. Predator occurrence in test arrays Spring 2017 experiment.**
(PDF)

**S2 Table. Predator occurrence in test arrays Summer 2017 experiment.**
(PDF)

**S3 Table. Predator occurrence in test arrays Summer 2018 experiment.**
(PDF)

## Acknowledgments

The authors wish to express their sincere gratitude to Diane Kimes, Executive Director of Heathcote Botanical Gardens, for providing access, encouragement, and logistical support, Nichol Gaza and Frank Manthey for their assistance with the fieldwork studies and sticky card trap analysis, Paul Robbins and William Meikle for reviewing an early draft of the manuscript, and three anonymous reviewers whose critical comments greatly improved the manuscript. Mention of trade names or commercial products in this publication is solely for the purpose of providing specific information and does not imply recommendation of endorsement by the USDA for its use.

## Author Contributions

**Conceptualization:** Joseph M. Patt, Randall P. Niedz.

**Data curation:** Aleena M. Tarshis Moreno.

**Formal analysis:** Joseph M. Patt, Randall P. Niedz.

**Investigation:** Joseph M. Patt, Aleena M. Tarshis Moreno.

**Methodology:** Joseph M. Patt, Aleena M. Tarshis Moreno, Randall P. Niedz.

**Project administration:** Joseph M. Patt.

**Supervision:** Joseph M. Patt.

**Writing – original draft:** Joseph M. Patt, Randall P. Niedz.

**Writing – review & editing:** Aleena M. Tarshis Moreno.

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
