## [Decision Letter · Decision Letter 0]

14 Nov 2019

PONE-D-19-26873

Developing a conservation biological control strategy for the Asian citrus psyllid: Response surface methodology reveals proportionality effects of plant species in conservation plantings

PLOS ONE

Dear Dr. Patt,

Thank you for submitting your manuscript to PLOS ONE. After careful consideration, we feel that it has merit but does not fully meet PLOS ONE’s publication criteria as it currently stands. Therefore, we invite you to submit a revised version of the manuscript that addresses the points raised during the review process.

This manuscript received mixed reviews from three very qualified reviewers. One has suggested a reject and cited some rather critical methodological and interpretive issues. The other two reviewers recommended a minor revision. However, they all expressed concerns and many were similar to the first reviewer. I am therefore suggesting that you revise this manuscript taking very particular care to address the common issues among the reviewers.

We would appreciate receiving your revised manuscript by Dec 29 2019 11:59PM. To enhance the reproducibility of your results, we recommend that if applicable you deposit your laboratory protocols in protocols.io, where a protocol can be assigned its own identifier (DOI) such that it can be cited independently in the future. For instructions see: http://journals.plos.org/plosone/s/submission-guidelines#loc-laboratory-protocols

We look forward to receiving your revised manuscript.

Kind regards,

Sean Michael Prager, Ph.D.

Academic Editor

PLOS ONE

Journal Requirements:

Reviewers' comments:

Reviewer's Responses to Questions

**Comments to the Author**

1. Is the manuscript technically sound, and do the data support the conclusions?

Reviewer #1: No

Reviewer #2: Yes

Reviewer #3: Yes

2. Has the statistical analysis been performed appropriately and rigorously? 

Reviewer #1: Yes

Reviewer #2: Yes

Reviewer #3: Yes

3. Have the authors made all data underlying the findings in their manuscript fully available?

Reviewer #1: Yes

Reviewer #2: Yes

Reviewer #3: Yes

4. Is the manuscript presented in an intelligible fashion and written in standard English?

Reviewer #1: Yes

Reviewer #2: Yes

Reviewer #3: Yes

5. Review Comments to the Author

Reviewer #1: Developing a conservation biological control strategy for ACP: response surface methods reveals proportionality effects of plant species in conservation plantings

Patt et al. seek to determine the optimal mixture of candidate insectary plants for conservation biological control of ACP. The study was extensive and completed over multiple years. The data collected was analyzed in an appropriate method. The biggest flaw of the paper is that it is unclear who the target audience of this work is-- is it homeowners, is it commercial citrus? The study was done in a location that would not reflect the communities in a citrus grove, a botanical garden-- a choice that is not justified in the manuscript. While the data is interesting, the authors do not sufficiently explain why such choices were made and who this information pertains to given the location of the study. Overall, the study provides general information on the blend of plants to a generic insect community that is outside the realm of commercial citrus production and Asian citrus psyllid.

Introduction:

Line 46- non-crop plant species doesn’t really make sense here. Especially as citrus, when it has floral resources is really attractive within the landscape. So, rephrase to something like "provided by enhanced plant diversity" -- something like this

47- "within ecologically impoverished landscapes of modern ecosystems- subjective -- correct to: Within monoculture agroecosystems,

89 -- It doesn’t seem from the literature that it has been a mixed result-- they have been overall, lackluster.

99-102-- would be better with a stronger argument as to why this method is worthwhile for a vector-driven pathosystem as biocontrol rarely removes all of the target insects. Insecticide resistance is good reasoning but maybe also something about Florida's situation being really high % of infection in groves so less interest in 100% control.

121: what is the argument for translatability between chemical composition and plant species composition-- make that argument here. Chemicals are stable components-- plants are living things that can vary in quality by nutritional status and stressors.

Table 1: add where the plants are native to-- this is of critical importance as it will allow people to choose species that relate best for their region if they have been tested in this study

227: was position of the box included in the analysis-- this could likely play a role of what was captured where--- since this was artificially placed in an outdoor arena.

-- is there any reason why the plants or boxes were not separated from eachother distinctly?

Table 3: might be missing something-- but is there a reason the plant mixtures were so different by season? How do you think this drives the differences by season?

M&M:

- Why did you choose this site? Does it have citrus? This is the biggest question I have about this experiment: why is ACP biological control being researched in a botanical garden which has significantly more plant diversity than most citrus production areas--- please justify? Or, maybe reframe the study completely as just conservation biological control for generalist predators. It is very unclear why this didn’t occur on the edge of a citrus grove.

- How were the plants cared for during the study-- what was the watering frequency?

- Why were sticky cards done without visual counts or some other additional specimen collection method?

Results:

- Why are there so many figures? Is there a logical way to lower this number? It is a distraction for the paper.

254: citation style has changed in this line

324: Honeydew producer is driving abundance-- ACP is also a HD producer. This result solidifies that the study is not well designed with regard to ACP. It could be reframed to not include ACP but then it might require going back through the cards in order to determine what other predators are present out of the scope of ACP. HD is a significant resource that ACP also provides and this resource drives various interactions between ants and ACP and other HD producers. It is likely that ACP are less attractive than the aphids present due to the waxy coating on their HD.

Reviewer #2: Developing a conservation biological control strategy for the Asian citrus psyllid: Response surface methodology reveals proportionality effects of plant species in conservation plantings.

Patt, Moreno & Niedz

Patt et al. investigate the effect of different proportions of nectar producing plant species on the abundance of predators of D. citri, an important pest of the citrus industry in Florida and California. They aimed to use conservation biocontrol plants that supply nectar and/or pollen to natural enemies of D. citri with the idea that these plants may attract and retain natural enemies of D. citri. However, there may be a few shortcomings of this study:

1. Three key natural enemies of D. citri were not sampled during the study – T. radiata, lacewings and hoverflies. Sticky traps are not effective at trapping hoverflies – they require pan traps or malaise traps. The study sampled coccinellids, predatory Hemiptera and a cockroach species which may or may not be a predator of D. citri egg masses.

2. Coccinellids have high mobility and likely dispersed well beyond the 3 m buffer zones between treatments. This study would have benefited from increased distance between treatments. Natural enemies were likely dispersing across many treatments, obtaining nectar and pollen from multiple treatments.

3. Yellow sticky traps were the only sampling method used, and these tend to trap natural enemies that require nectar by mimicking the yellow color of a flower and attracting hungry individuals. A yellow sticky trap placed beside white flowering buckwheat may not trap natural enemies that prefer feeding from white flowers. A yellow sticky trap surrounded in bare ground in the control will ‘stand out’ to hungry natural enemies and be more attractive than a sticky trap surrounded by, and competing with, flowers.

4. Instead of investigating the effect of nectar producing plants on abundance of predators, the study more likely determined the effect of different proportions of aphid host plants on the abundance of predators (ie., plants that aphids prefer and that subsequently effected predator abundance). For example, assassin bugs have piercing mouth parts that penetrate prey and suck out their juices so they don’t feed on nectar and pollen.

5. Page 21: Line 396: A large body of research has shown the benefits of buckwheat in conservation biocontrol – this study likely did not find a significant effect of buckwheat on predator abundance due to the species of natural enemies sampled, small buffer zones between treatments, and the one trapping method used.

6. This study would have benefited from an additional sampling method to capture other key natural enemies of D. citri, and timed observations of flowers/extrafloral nectaries to determine whether natural enemies were actually using nectar/flower resources. Sticky traps “attract” natural enemies which may led to bias, whereas vacuum sampling, sweep netting, malaise traps or timed observations are methods that do not use a trapping mechanism.

Despite the potential shortcomings, this study is a helpful first “proof of concept study” and demonstrates that a combination of three plant species provided a good habitat for some natural enemies of D. citri. It also raises the possibility of using banker plants to supply alternative hosts for natural enemies of D. citri, however, it is paramount to determine whether natural enemies from such banker systems disperse into the orchard for biocontrol. The manuscript has a large number of tables and figures. Suggest combining Tables 3, 4, and 5 into one table? Figs 5-7 could be combined into 2 figures - the triangle part of Figs. 5-7 could be combined into one figure (a-d), and the deviation from reference point parts of Fig. 5-7 could be combined into one figure (a-d). The manuscript is well written and prepared. A few minor suggestions are as follows:

Page 4, line 73: Should this read “…..literature on the efficacy….”?

Page 5, Line 93: Change impacts to impact?

Page 12, Line 231: What height were the sticky traps place?

Fig 9: The axis is labelled “no. combined predators” rather than coccinellids.

Page 23, Line 426: Cite references?

Page 23, Line 432: Was the aphid infestation measured or observed?

Page 24, Line 461: It is unknown whether the hemipteran predators sampled would disperse from banker plants into the orchard for control of D. citri.

Page 25: Line 480-485: Parasitoids of D. citri were also not sampled. Sticky traps are not an effective way to captured hoverflies – they require malaise or pan traps. Suggest to cite references that have investigated what are the key natural enemies of D. citri in FL and what proportion of D. citri each natural enemy attacks. Conducting this study among or near citrus orchards may have helped capture the desired natural enemies.

Recommendation: Accept Submission with updated discussion about the shortcomings of this study and further study needed using additional sampling techniques, timed observations of natural enemies feeding on nectar sources, and increased buffer zones between treatments.

Reviewer #3: Title: Developing a conservation biological control strategy for the Asian citrus psyllid […]

This is a very interesting article. The authors use RSM to establish the best mix of flowering plants to attract natural enemies. Adding flowers resource to agrosystems in order to provide food resource to natural enemies has been studied for a long time in conservation biological control. However, this is the first time, to my knowledge, that the surface method is used to address this problem. I think this is a brilliant idea that by itself justifies the publication of the article in PLoS One. On the top of that, there are some valuable information regarding flowers to use to increase natural enemies of Diaphorina citri in Florida. As most of the species recorded are generalist predators the information provided could be used to increase natural enemies in other agrosystems of Florida.

MAJOR COMMENT

There is one limitation in this study that need to be discussed

- The authors used sticky traps data as the primary response variable. This has two constraints: first, sticky traps capture insects that did not necessarily feed or were present in the selected plant. To be captured on a sticky trap, the insect has to move from one plant to another. This is an indirect assessment of insect diversity. Second, more than the diversity what we are really looking for is the effect on D. citri population. As a follow up experiment, we could imagine having some psyllid-infested citrus plants within these flower beds and compare D. citri populations over time. There are multiple experiments that have been conducted showing that flower strips can increase predator densities, but it does not necessarily transfer into a control of the prey.

MINOR COMMENTS

L24: replace by “two times a week”

L33-35: I think the authors should mention which plants had linear effects on predator abundance. This is an information that most of the readers would like to have when reading the abstract.

L38-39 To me the main output from this article is that a single species may drive the abundance of predator. Therefore, the question of having complex mix of flower might be questioned.

L57: insert the Latin name for Coriander.

L80: change to […] biocontrol of Diaphorina citri Kuwayama (Hemiptera: Liviidae), the vector of the causal agent of …

L81 maybe use the widely accepted acronym HLB for the article

L95 change to “multiple insecticide classes”

L209: it was not very clear to me which treatments listed table 2 the ‘model’ and ‘lack of fit’ points referred too.

L215: I believe the control treatment (0,0,0) should be listed in table 2. Also, it should be displayed in Fig. 2

L216: I am not sure if the planter boxes of the same treatment were side by side or separated as it is suggested Fig. 2. If there was a space between planter boxes within the same treatment this should be mentioned in the text and in the caption of fig. 2

L232: Did the authors corrected their data for the slight difference in duration? The data could be expressed as insect/trap/day.

L246: I never read that Blatella asachina is a predator of psyllids, I only think about it as a Lepidopetra egg predator. Maybe change to ‘potential predator’ unless the authors observed predation of psyllid by B. asachina?

Tables 3, 4, 5: it is mentioned that there were 2 arrays of control, whereas L215 it is only one.

Table 3: I found a little bit confusing that the number of arrays was only mentioned for Spring 2017. For clarity I would also add it to summer 2017 and summer 2018. Same thing for tables 4 and 5.

Fig4-7: I believe that it will be easier if the 4 figures were all combined in a single one with 8 different panels. Same thing for figures 9 and 10.

Table 6: I am wondering if it’s normal that we do not see the individual linear response (A, B, and C) of each plant in the model (they are all included in the ‘linear mixture’). This is not too much a trouble as the trace plots are a nice visualization of the individual effect of each plant. Nevertheless, it is weird to have interactions showing up but not the individual effect. Maybe a little explanation would be nice.

Table6: precise that non significant interactions (α=0.05?) were removed from the model.

L459-461: it seems that to make this conclusion we should have non-infested P. lunatus.

6. PLOS authors have the option to publish the peer review history of their article (what does this mean?). If published, this will include your full peer review and any attached files.

Reviewer #1: No

Reviewer #2: No

Reviewer #3: No

---

## [Author Response · Author response to Decision Letter 0]

24 Feb 2020

Dear Dr. Prager:

Please accept for your consideration the following rebuttal letter to the Reviewers of our manuscript PONE-D-19-26873, ‘Developing a conservation biological control strategy for the Asian citrus psyllid: Response surface methodology reveals proportionality effects of plant species in conservation plantings’.

I have made all of the changes suggested by the Reviewers except in those cases where I disagreed with the Reviewer. In these instances, I provided an explanation for why I disagreed with the Reviewer and, if warranted, modified the manuscript to provide clarification in cases where the points I was trying to make in the initial submission were unclear. 

All of the changes made in the text are marked in blue font or strikeout. 

Because all of the Reviewers were concerned about the number of figures, I have combined Figures 4 to10 into two figures as suggested.

I wish to thank all of the Reviewers for their comments and suggestions as these have significantly improved the manuscript. I greatly appreciate the time and effort you all have made on our behalf.

Best Regards—Joe Patt

Reviewer #1. 

Comment 1: The biggest flaw of the paper is that it is unclear who the target audience of this work is-- is it homeowners, is it commercial citrus?

Because this was a proof-of-concept study, the primary audience of this paper will be researchers, particularly those working on biological control, IPM, formulation of mixtures, and design of experiments. Ultimately our goal will be to devise an explicit set of recommendations with a list of plants that can support biocontrol agents of D. citri in either residential landscapes or commercial citrus groves and the best means to grow them; i.e., as linear strips along the grove border, as a cover crop, etc.. However, further evaluations are needed in each of those particular environments to do so and we expect that the methodology we present here will contribute to this effort. We alluded to this in the Introduction: L.102: ‘This may provide an opportunity to develop and implement effective conservation biological control strategies to suppress D. citri populations in commercial citrus groves and nearby residential and commercial landscapes.’

To clarify and re-enforce the points that this was a proof-of-concept study whose results were intended as a building block for developing conservation biological control strategies for D. citri in different landscapes, I added the following language (in blue font) to the manuscript: 

In the Introduction:

L. 42: The addition of certain plant species to depauperate landscapes can provide significant resource subsidies to predaceous and parasitic arthropods, and may result in increased suppression of pest insects. 

L105: As a first step in developing a conservation biological control strategy for D. citri, we compared the efficacy of single-species versus mixed species plantings of nectary and banker plants (‘conservation plants’) with respect to supporting the psyllid’s predators. 

In the Discussion:

L503: Potentially, a variety of nectary and banker plant species could be used to support natural enemies of D. citri. Selection of conservation plants species will depend, in large part, on the target landscape where psyllid suppression is wanted. For example, aesthetic appeal will be an important attribute for conservation plants grown in residential landscapes while those grown in citrus groves may need to be drought tolerant and amenable to mechanized planting and cultivation. The purpose of the present study was to demonstrate that RSM could be used as a tool to evaluate the effectiveness of different plant mixtures with respect to the occurrence of the psyllid’s predators. This information will permit us to select plant species or species mixtures that are optimal for promoting conservation biological control in residential landscapes, commercial groves, or other targeted landscapes. 

Comment 2: The study was done in a location that would not reflect the communities in a citrus grove, a botanical garden-- a choice that is not justified in the manuscript.

Since this was a proof-of-concept study, we felt that it needed to be conducted at an appropriate and manageable scale. We chose a site that was reflective of the Indian River region in general, could supply irrigation water, was secure, and was not treated with insecticides—a factor that could have severely impacted the results of the study. Commercial citrus growers in our area frequently spray insecticides, sometimes without any prior warning to cooperators, which disqualifies commercial groves as being a suitable location for a proof-of-concept study. Heathcote Botanical Gardens met all the requirements listed above. 

I have added language to the text to clarify this:

(L151): 

Because this was a proof-of-concept study, we wanted to conduct the tests at an appropriate and manageable scale. We chose Heathcote Botanical Gardens, a 2.8-hectare public display garden located in Fort Pierce, FL as the study site. Formerly the site of a commercial ornamental nursery, the vegetation of Heathcote Botanical Gardens is representative of residential areas and the urban/farmland interface in our region. The site was secure, could provide water, and, importantly, was not treated with insecticides, which would have confounded the results. The study site itself was located in an open, mown area periodically used as a parking lot and exposed to full sun throughout the day. It was surrounded by mature trees, primarily southern live oak (Quercus virginiana Mill.). The open conditions at the study site were representative of residential areas and citrus grove borders.

Because Heathcote was formerly an ornamental plants nursery, the plants on site are typical of residential landscapes in southern Florida. Immediately across the street from the study site is the US Route 1 corridor, comprised of commercial sprawl. Below is a screen shot of Google Maps showing Heathcote Botanical Garden (outlined in white), with the study site indicated by a red star, and showing the adjoining neighborhoods and commercial zone. 

For context, the screen shot below shows the location of a 35-acre commercial citrus grove (indicated by a red star) in the Indian River citrus production area. We conducted a preliminary investigation at that grove that lead to the study conducted at Heathcote Botanical Gardens. The commercial grove is adjoined by expansive condominium complexes and is near a large shopping mall. Nearby disturbed areas are occupied by thick stands of exotic invasive plants such as Brazilian pepper. The diversity of coccinellids collected from Heathcote and the commercial citrus grove were similarly dominated by exotic species such as Harmonia axyridis. The high degree of urban encroachment on farmland (at least in our area) makes moot the Reviewer’s comment about the study was done in a location that would not reflect the communities in a citrus grove.

Comment 3: While the data is interesting, the authors do not sufficiently explain why such choices were made and who this information pertains to given the location of the study.

I’m not clear about what the Reviewer meant by ‘why such choices were made’ in the comment. If it is question of which plant species were selected for the study, the rationale was stated in the manuscript: (L164-8): ‘Each of these plant species met the criteria for the ‘right kind of plants’ for biological control agents as described by Wäckers and van Rijn [32] in that they had either extrafloral nectaries or open shallow flowers whose nectaries were accessible to natural enemies such as T. radiata which possess short mouthparts and a limited floral foraging repertoire [74].’ 

We selected plant species with exposed nectaries because according to multiple studies, including our own, flowers with these types of nectaries should attract and support predaceous arthropods and as this is a basic ecological interaction it should occur regardless of whether in a grove or a backyard. As mentioned above, the species of plants that are suitable and optimal for each environment needs to be determined, but that is beyond the scope of the present study. We hope to address this in future studies.

The second part of comment 3 was addressed in the previous responses.

Comment 4: Overall, the study provides general information on the blend of plants to a generic insect community that is outside the realm of commercial citrus production and Asian citrus psyllid.

I respectfully disagree. This was a proof-of-concept study of the RSM methodology, not one that describes a study further along in the development of the approach and that provides plant species recommendations for a particular environment (grove vs. residential). This study represents a first step in that direction. Also, as the maps above show, commercial citrus groves here and in many areas of Texas and California are near urban development. Several studies have shown that D. citri readily moves between residential areas and commercial citrus. So, to state that this study is ‘outside the realm’ of commercial citrus production and Asian citrus psyllid is not factual. This is south Florida in the Anthropocene.

Line 46- non-crop plant species doesn’t really make sense here. Especially as citrus, when it

has floral resources is really attractive within the landscape. So, rephrase to something like

"provided by enhanced plant diversity" -- something like this. 

I respectfully disagree. While citrus flowers are a good resource for bees, sweet orange and grapefruit trees bloom for only 10 to 14 days, making them a short-lived resource. In addition, the nectaries of citrus flowers are surrounded by a stiff wall of filaments, which in some citrus types are fused, making them very difficult to access by predaceous arthropods with short mouthparts. I’ve inserted the word ‘certain’ before ‘non-crop plants’ to make clear that it is only particular plant species are useful for providing nutritional resources to biocontrol agents.

47- "within ecologically impoverished landscapes of modern ecosystems- subjective -- correct

to: Within monoculture agroecosystems. Agree. Done.

89 -- It doesn’t seem from the literature that it has been a mixed result-- they have been

overall, lackluster. I agree but a California researcher recently told me that they are seeing more promising results, so I am opting to keep the language the same.

99-102-- would be better with a stronger argument as to why this method is worthwhile for a

vector-driven pathosystem as biocontrol rarely removes all of the target insects. Insecticide

resistance is good reasoning but maybe also something about Florida's situation being really

high % of infection in groves so less interest in 100% control.

I agree with the Reviewer’s sentiment. However, in Florida, the growers themselves are cutting back on insecticidal control because it has not stopped the spread of HLB and they are opting instead to spend their money on anti-microbials, fertilizers and foliar nutrient sprays. Anecdotal evidence suggests that natural enemies are returning to the groves as a consequence. Given the increasing cost of pest control and fertilizers, growers are beginning to consider biological control as an option to cut costs. Some growers are beginning to grow ground covers and are seeing enhanced tree health and fruit production from trees that were sickly and are now reinvigorated (citrusindustry.net/2019/11/18/cover-crops-bring-hlb-recovery/ ). In the environment of HLB, growers are learning that investing in tree health helps with productivity. I’ve made an insertion in the text to show that financial considerations by growers is providing an opportunity to evaluate the conservation biological control as a feasible pest control strategy in the environment of HLB: 

L96: Rather than depend on insecticidal control of D. citri, commercial citrus growers in Florida are opting to purchase other measures such as antimicrobial applications to suppress CLas [65-66] and enhanced fertilization and foliar nutritional sprays to enhance tree health.

L101: As chemical suppression is reduced, natural enemies are expected to return to citrus groves, and growers are beginning to consider biological control as an option to cut costs. This may will provides an opportunity to develop and implement effective conservation biological control strategies to suppress D. citri populations in commercial citrus groves and nearby residential and commercial landscapes. 

121: what is the argument for translatability between chemical composition and plant species

composition-- make that argument here. Chemicals are stable components-- plants are living

things that can vary in quality by nutritional status and stressors.

I appreciate the Reviewer’s concern. As this was a proof-of-concept test, we did not know a priori if RSM would work with mixtures of plant species. As far as we are aware, no one has tried using RSM before with this kind of systems. We included the references about the pheromone component, diet mixture studies, etc. to provide the reader with background information about RSM and its application in entomological systems. I respectfully disagree with the reviewer’s suggestion about the need to make an argument about translatability between molecules and organisms as I feel that is beyond the scope of the study. The point is that the RSM approach worked with plant components and our results suggest that it is useful means of optimizing plant mixtures for the purpose of attracting and sustaining insect predators. We standardized the system as much as possible in that we used planter boxes of the same dimensions, same watering and maintenance regimen and keeping planting densities consistent. 

Table 1: add where the plants are native to-- this is of critical importance as it will allow

people to choose species that relate best for their region if they have been tested in this study.

The table already says which two plant species are native; this material is also provided in the text. I have added the descriptors ‘ornamental’ and ‘garden’ to the table for clarification.

227: was position of the box included in the analysis-- this could likely play a role of what was

captured where--- since this was artificially placed in an outdoor arena.

-- is there any reason why the plants or boxes were not separated from each other distinctly?

I appreciate the Reviewer’s concern. The planter boxes in each group were not separated from each other because we were testing the effect of a mixture of plant species on target insect occurrence. Separating the planter boxes from each other would defeat the purpose of the experiment because they would no longer be mixtures. The experimental design included ‘pure blends’, groups comprised only of a single species to test the effect of a single plant species on target insect occurrence. Because two trap cards were placed along the center line of each group, it is not likely that the placement of a particular species within the group affected the outcome. Also, the position of a particular species was not the same from test group to test group, reducing the likelihood of a position effect. I have modified Figure 2 so that the groupings of the planter boxes is clearer.

Table 3: might be missing something-- but is there a reason the plant mixtures were so

different by season? How do you think this drives the differences by season?

The plants were selected on the basis of their ability to attract and sustain insect predators and to grow well in the hot climate of south Florida. I have added the following language to clarify this point:

L 186: Added the following language: ‘All of the plant species grow well and readily flower in south Florida and are easily cultivated there during the warm season.’

L. 190: Drought tolerant species included E. milii, P. umbricola, and C. fasciculata, which, along with F. esculentum, were used in the experiments during the summer. 

In terms of methodology, we could have used the same three plant species for all three experiments, but we wanted to test a range of species to make sure that RSM could distinguish differences among several different mixtures of plant species. In other words, we felt that having different mixtures of plants in three different experiments was a more robust approach for testing RSM than using the same plant species mixture in all three experiments. Our point was not to figure out seasonal differences, which is a topic for further study all by itself.

To further clarify the point that we are not sure why we observed seasonal differences, the following language was added to the text: L. 454: ‘Since this was a proof-of-concept study, we do not know why nonlinear blending effects were not observed in the two summer experiments. Possible explanations could be because of a low ambient levels of predators, due to the summer heat, or to some other factor(s) attributable to the plant species used in each mixture experiment. 

This is followed by further discussion:

‘Another possibility is that, at low predator densities, larger plant mixture arrays are required to reveal nonlinear blending effects. Blaauw and Isaacs [79] found that natural enemy density, group richness, and diversity was greater in larger wildflower test plots than in smaller ones. A different experimental design, where the same plant mixtures are tested across seasons and with differently sized mixture arrays is needed to adequately explore this issue.’

M&M:

Why did you choose this site? Does it have citrus? This is the biggest question I have about

this experiment: why is ACP biological control being researched in a botanical garden which

has significantly more plant diversity than most citrus production areas--- please justify? Or,

maybe reframe the study completely as just conservation biological control for generalist

predators. It is very unclear why this didn’t occur on the edge of a citrus grove.

Please see response to comment 2. 

How were the plants cared for during the study-- what was the watering frequency?

Inserted into text: L 205. …maintain soil moisture and prevent weeds. Plants were watered as needed, typically once or twice per week depending on the frequency of summer thunderstorms.

Please not that in the text we stated that we used standard commercial potting soil and added a slow release fertilizer to the planter boxes at the start of each experiment.

Why were sticky cards done without visual counts or some other additional specimen

collection method?

All of the Reviewers had this problem with our methods and I can only plead mea culpa. Since this was a preliminary study, I felt that a single collection method was sufficient, and indeed the single collection method worked in terms of demonstrating the utility of RSM in this system. Another reviewer thought we should have used water-pan traps, especially to capture syrphids. We tried using them, but the high frequency of rainstorms in our area makes their use untenable as they overflowed from rainwater and the soap solution became too diluted. In our subsequent studies (not yet published), we have included standardized visual inspections in addition to the card traps. These standardized visual inspections have verified the data from the card traps in the present study by showing that syrphid and predatory hemipteran diversity and abundance is low at the site. We have not detected any difference between visual inspections and card trap captures with respect detecting diversity of large-bodied coccinellid species, and visual inspection doesn’t permit us to detect small-bodied coccinellids with any confidence. However, the visual inspections are showing that the abundance of large-bodied coccinellids is greater than what is indicated by the card trap captures, but only on the plants supporting aphids. We have not yet conducted nighttime visual observations to ascertain their utility with respect to Blattella asahinai; however, they are captured in high numbers on the card traps.

Results:

Why are there so many figures? Is there a logical way to lower this number? It is a

distraction for the paper.

Another reviewer commented on this and I have combined Figures 4-10 into 2 figures.

254: citation style has changed in this line. Corrected.

324: Honeydew producer is driving abundance-- ACP is also a HD producer. This result

solidifies that the study is not well designed with regard to ACP. It could be reframed to not include ACP but then it might require going back through the cards in order to determine what other predators are present out of the scope of ACP. HD is a significant resource that ACP also provides and this resource drives various interactions between ants and ACP and other HD producers. It is likely that ACP are less attractive than the aphids present due to the waxy coating on their HD.

I respectfully disagree with the statement that ‘the study in not well designed with regard to ACP’ because it produces honeydew and that ‘This result solidifies that the study is not well designed with regard to ACP.’. Banker plants provide predators with alternate prey (which typically produces HD) and that the predators then attack the crop pest (which also typically produce HD). Fluctuations in psyllid populations are driven primarily by the amount of flush available to them. Maintaining sources of alternate prey, pollen and nectar within or near a grove should support predator populations when psyllid populations are low. The current study was a first step in showing that the psyllid’s predators occur on plants that provide them with nutritional resources, and that by modifying the proportion of plants within the group, we could affect the outcome with respect to predator occurrence. Now that we have a grasp of which plant species are suitable for supporting psyllid predators, we will move ahead and determine whether planting them results in significantly higher mortality of psyllids in nearby citrus. 

Reviewer #2: 

Developing a conservation biological control strategy for the Asian citrus

psyllid: Response surface methodology reveals proportionality effects of plant species in

conservation plantings.

Patt, Moreno & Niedz

Patt et al. investigate the effect of different proportions of nectar producing plant species on

the abundance of predators of D. citri, an important pest of the citrus industry in Florida and

California. They aimed to use conservation biocontrol plants that supply nectar and/or pollen

to natural enemies of D. citri with the idea that these plants may attract and retain natural

enemies of D. citri. However, there may be a few shortcomings of this study:

1. Three key natural enemies of D. citri were not sampled during the study – T. radiata,

lacewings and hoverflies. Sticky traps are not effective at trapping hoverflies – they require

pan traps or malaise traps. The study sampled coccinellids, predatory Hemiptera and a

cockroach species which may or may not be a predator of D. citri egg masses.

We searched for, but did not find, Tamarixia on the trap cards. We tried to use pan traps but the high frequency of thunderstorms in our area precluded their use because they overflowed and the soap solution became diluted. Since this study was completed, we have added repetitive, standardized visual observations as a means of assessing predator occurrence in the test plots. Based on our visual observations, taken in mid-morning, syrphid abundance and diversity at the site appears to be much lower than that of coccinellids. 

2. Coccinellids have high mobility and likely dispersed well beyond the 3 m buffer zones

between treatments. This study would have benefited from increased distance between

treatments. Natural enemies were likely dispersing across many treatments, obtaining nectar

and pollen from multiple treatments.

I acknowledge the reviewer’s concerns. We were restricted from increasing the gap between treatment groups because of the size of the study site available to us. However, I would like to point out that Blaauw and Isaacs conducted a study to test the effect of wildflower plot size on natural enemy abundance and biological control activity and their test plots were fairly close to each other (Ecological Entomology 37: 386-394 2012). An aerial view of their study site is shown below.

Fig. 1. Aerial image of wildflower plantings in August 2010. Sized plots were arranged using a Latin-square design, and the size (m2) of the wildflower plantings in the left column are displayed at the side of the photo.

3. Yellow sticky traps were the only sampling method used, and these tend to trap natural

enemies that require nectar by mimicking the yellow color of a flower and attracting hungry

individuals. A yellow sticky trap placed beside white flowering buckwheat may not trap

natural enemies that prefer feeding from white flowers. A yellow sticky trap surrounded in

bare ground in the control will ‘stand out’ to hungry natural enemies and be more attractive

than a sticky trap surrounded by, and competing with, flowers.

I acknowledge the reviewer’s point about the yellow sticky card traps. Also, I inadvertently left out a detail about the ‘control group’. To prevent the yellow sticky card traps from ‘standing out’, we placed a burlap curtain around the perimeter of the planters to approximate the level of visibility of the trap cards in the control to that in the planted groups. The burlap was loosely-woven and the cards were visible through it. I have added the following detail to the text: L.216 A curtain made of loosely-woven burlap, extending upwards 1M from the top of the planter boxes, and supported by bamboo poles, was placed around the perimeters of the two control groups to approximate the visibility of the yellow sticky card traps in the interior of the planted groups. 

4. Instead of investigating the effect of nectar producing plants on abundance of predators, the

study more likely determined the effect of different proportions of aphid host plants on the

abundance of predators (ie., plants that aphids prefer and that subsequently effected predator

abundance). For example, assassin bugs have piercing mouth parts that penetrate prey and

suck out their juices so they don’t feed on nectar and pollen.

I agree with the reviewer. In the Discussion, we stated: “There were several unexpected outcomes of the study. The first was a banker plant effect, observed in the Spring 2017 experiment, in which an aphid infestation on P. lunatus was the primary driver of predator abundance. Our original concept was that plant-provided nutrients would drive predator occurrence in the test mixtures and P. lunatus was selected for the experiment because it has stipular extrafloral nectaries. The results here indicate that banker plant species that can support non-pestiferous alternate prey may be an important component of plant mixtures grown to support natural enemies of D. citri, as is the case in other systems [5,80].” 

5. Page 21: Line 396: A large body of research has shown the benefits of buckwheat in

conservation biocontrol – this study likely did not find a significant effect of buckwheat on

predator abundance due to the species of natural enemies sampled, small buffer zones between

treatments, and the one trapping method used.

I am perplexed as to why we did not collect more predators from the buckwheat. I agree with the reviewer about the sampling methods, etc. We may not have seen more predators on the buckwheat because they were outcompeted by swarms of scavenger flies, which probably came from the dumpster of a restaurant adjacent to the study site (Location of dumpster is shown by red arrow in screen shot of study site).

6. This study would have benefited from an additional sampling method to capture other key

natural enemies of D. citri, and timed observations of flowers/extrafloral nectaries to

determine whether natural enemies were actually using nectar/flower resources. Sticky traps

“attract” natural enemies which may led to bias, whereas vacuum sampling, sweep netting,

malaise traps or timed observations are methods that do not use a trapping mechanism.

Despite the potential shortcomings, this study is a helpful first “proof of concept study” and

demonstrates that a combination of three plant species provided a good habitat for some

natural enemies of D. citri. It also raises the possibility of using banker plants to supply

alternative hosts for natural enemies of D. citri, however, it is paramount to determine whether

natural enemies from such banker systems disperse into the orchard for biocontrol. 

I agree with the Reviewer’s comments. During this growing season, we will measure psyllid mortality in sentinel citrus at increasing distances from the planter boxes. We are also examining coccinellid frass pellets to assess the amounts and types of pollens ingested as well as the abundance of prey exoskeleton fragments.

The manuscript has a large number of tables and figures. Suggest combining Tables 3, 4, and 5

into one table? Figs 5-7 could be combined into 2 figures - the triangle part of Figs. 5-7 could

be combined into one figure (a-d), and the deviation from reference point parts of Fig. 5-7

could be combined into one figure (a-d). 

Other reviewers commented about the number of figures and I have combined Fig 4-10 into 2 figures.

The manuscript is well written and prepared. A few minor suggestions are as follows:

Page 4, line 73: Should this read “…..literature on the efficacy….”? Corrected.

Page 5, Line 93: Change impacts to impact? Corrected.

Page 12, Line 231: What height were the sticky traps place? Inserted into text: ‘100 cm above the ground…’

Fig 9: The axis is labelled “no. combined predators” rather than coccinellids. Corrected.

Page 23, Line 426: Cite references? The statement is our speculation and I am not aware of a reference to cite for it.

Page 23, Line 432: Was the aphid infestation measured or observed? Unfortunately, we did not measure aphid infestation level, only to say that the bean plants were ‘heavily infested’. I estimate that, collectively, the bean plants had an aphid population numbering in the thousands.

Page 24, Line 461: It is unknown whether the hemipteran predators sampled would disperse

from banker plants into the orchard for control of D. citri. We will conduct a test this year to determine whether this is so.

Page 25: Line 480-485: Parasitoids of D. citri were also not sampled. Sticky traps are not an

effective way to captured hoverflies – they require malaise or pan traps. Suggest to cite

references that have investigated what are the key natural enemies of D. citri in FL and what

proportion of D. citri each natural enemy attacks. The ms already has citations to Qureshi & Stansly (2009) and the studies conducted by J. P. Michaud; these are the baseline studies on natural enemies of psyllids in Florida. While these studies describe the frequency of natural enemies sampled, I am not aware of any studies that have determined what proportion of psyllids are attacked by different types of natural enemies in Florida. 

Conducting this study among or near citrus orchards may have helped capture the desired natural enemies. See comments above to Reviewer 1.

Recommendation: Accept Submission with updated discussion about the shortcomings of this

study and further study needed using:

A. additional sampling techniques. Added to L.460: The yellow sticky card traps may have been ineffectual in capturing these insects and future studies should include additional sampling methods such as visual inspections and malaise traps. Pan traps were not included as a sampling method because frequent warm season thunderstorms caused them to overflow and precluded their use.

B. timed observations of natural enemies feeding on nectar sources. Added to L.492: These studies should include additional measures, such as a timed observation component and gut content analysis, to determine the level of feeding by each predator group on the different nutritional resources (pollen, nectar, alternate prey) present within groups of test plants.

C. increased buffer zones between treatments. Added to L. 497: We will also determine the importance of plant group size and planting arrangements in attracting and supporting natural enemies, especially during periods of low D. citri abundance and at different seasons of the year. For example, studies will determine whether a single strip of insectary plants placed along the grove border is as effective in suppressing psyllids as other types of planting arrangement, such as multiple strips planted at intervals throughout the grove or a cover crop grown across the grove understory.

Reviewer #3: 

Title: Developing a conservation biological control strategy for the Asian citrus

psyllid […]

This is a very interesting article. The authors use RSM to establish the best mix of flowering

plants to attract natural enemies. Adding flowers resource to agrosystems in order to provide

food resource to natural enemies has been studied for a long time in conservation biological

control. However, this is the first time, to my knowledge, that the surface method is used to

address this problem. I think this is a brilliant idea that by itself justifies the publication of the

article in PLoS One. On the top of that, there are some valuable information regarding flowers

to use to increase natural enemies of Diaphorina citri in Florida. As most of the species

recorded are generalist predators the information provided could be used to increase natural

enemies in other agrosystems of Florida.

MAJOR COMMENT

There is one limitation in this study that need to be discussed

- The authors used sticky traps data as the primary response variable. This has two constraints:

first, sticky traps capture insects that did not necessarily feed or were present in the selected

plant. To be captured on a sticky trap, the insect has to move from one plant to another. This is

an indirect assessment of insect diversity. Second, more than the diversity what we are really

looking for is the effect on D. citri population. As a follow up experiment, we could imagine

having some psyllid-infested citrus plants within these flower beds and compare D. citri

populations over time. There are multiple experiments that have been conducted showing that

flower strips can increase predator densities, but it does not necessarily transfer into a control

of the prey.

Other reviewers have criticized us for using only a single sampling method and I have addressed this shortcoming in my responses above and in the text. In the approaching growing season, we are planning to determine whether growing these plants has an effect on D. citri mortality in nearby host plants.

MINOR COMMENTS

L24: replace by “two times a week” Corrected.

L33-35: I think the authors should mention which plants had linear effects on predator

abundance. This is an information that most of the readers would like to have when reading

the abstract. 

Text added to Abstract L.33: In both Summer experiments, only linear mixture effects were observed, indicating that predator occurrence was driven by the amount of a single plant species in the test mixtures: Euphorbia milii in 2017 and Portulaca umbricola in 2018.

L38-39 To me the main output from this article is that a single species may drive the

abundance of predator. Therefore, the question of having complex mix of flower might be

questioned. I see the Reviewer’s point, though I don’t agree 100% with it. It was true in the two summer experiments. Interestingly, though, we did see a mixture effect in the Spring 2017 Experiment where the occurrence of predators was lower in pure mixture of lima bean than in test group comprised of 2/3 lima bean and 1/3 the other two species. This is shown in Figure 4g and Figure 3. Some Florida citrus growers have resurrected unproductive, HLB-infected groves by planting a mixture of ground cover plants. RSM may be useful in developing optimal ground cover mixtures.

L57: insert the Latin name for Coriander. Corrected.

L80: change to […] biocontrol of Diaphorina citri Kuwayama (Hemiptera: Liviidae), the

vector of the causal agent of … Corrected.

L81 maybe use the widely accepted acronym HLB for the article. Corrected.

L95 change to “multiple insecticide classes”. Corrected.

L209: it was not very clear to me which treatments listed table 2 the ‘model’ and ‘lack of fit’

points referred too. Corrected.

L215: I believe the control treatment (0,0,0) should be listed in table 2. Also, it should be

displayed in Fig. 2. I have included the following information in the Table 2 legend: Note that the ‘control’ treatment (0,0,0) is not included here because there is no null mixture in the design space; it was included in the experiments only as a reference. I have added the control treatment groups to Figure 2.

L216: I am not sure if the planter boxes of the same treatment were side by side or separated

as it is suggested Fig. 2. If there was a space between planter boxes within the same treatment

this should be mentioned in the text and in the caption of fig. 2. Figure 2 was redrawn to clarify the placement of the individual planter boxes within each treatment group. A photograph showing a section of the test array with a number of treatment groups has been added to Figure 2 to help the reader understand the placement of the individual planter boxes and the arrangement of the treatment groups.

L232: Did the authors corrected their data for the slight difference in duration? The data could

be expressed as insect/trap/day. For clarification, the following statement was inserted in the text: L.241. The specimen data from each collection period performed during a single experiment (Spring 2017, Summer 2017, Summer 2018) were pooled for analysis. 

L246: I never read that Blatella asachina is a predator of psyllids, I only think about it as a

Lepidopetra egg predator. Maybe change to ‘potential predator’ unless the authors observed

predation of psyllid by B. asachina? Qureshi and Stansly (2009) found a high number of B. asahinai on sticky barriers near psyllid colonies but did not conduct the necessary nocturnal observations to ascertain that they were preying on psyllids. We have clarified this point in the text.

Tables 3, 4, 5: it is mentioned that there were 2 arrays of control, whereas L215 it is only one. Corrected.

Table 3: I found a little bit confusing that the number of arrays was only mentioned for Spring

2017. For clarity I would also add it to summer 2017 and summer 2018. Same thing for tables

4 and 5. Corrected.

Fig4-7: I believe that it will be easier if the 4 figures were all combined in a single one with 8

different panels. Same thing for figures 9 and 10. Figures were combined.

Table 6: I am wondering if it’s normal that we do not see the individual linear response (A, B,

and C) of each plant in the model (they are all included in the ‘linear mixture’). This is not too

much a trouble as the trace plots are a nice visualization of the individual effect of each plant.

Nevertheless, it is weird to have interactions showing up but not the individual effect. Maybe a

little explanation would be nice.

Very good observation. Yes, it is normal to only include the “linear model” rather than the individual components in a mixture ANOVA. The reason is that the individual components are considered together (compared to each other) because they are not independent. The linear coefficients are estimates of the response at each vertex – not estimates of the effects of the components. Unfortunately, some software packages label the coefficients as “effects”. The linear coefficients are not shown to avoid this confusion. The effect of a mixture component is defined by a gradient (or slope) in some specified direction. The trace plot nicely captures the effects of the components. The “interaction” terms also have to be interpreted differently in a mixture analysis. What look like interaction terms are called quadratic blending or nonlinear blending terms and show if the two components are exhibiting a synergism or antagonism – i.e., a response not predicted by the simple additive blending in the linear mixture model.

To address the Reviewer’s concerns, the legend for Table 6 has been re-written as follows: Reduced quadratic response surface model, including diagnostic statistics, for natural enemy occurrence in mixture arrays containing three plant species. The “linear model” term is used in a mixture ANOVA because the individual components are not independent and are considered together (compared to each other). The effect of a mixture component is defined by a gradient (or slope) in some specified direction, and these effects are shown in the trace plots (Figs 4 & 5). The terms AB, AC, and BC are quadratic blending or nonlinear blending terms in mixture models and show if the two components are exhibiting a synergism or antagonism – i.e., a response not predicted by the simple additive blending in the linear mixture model. They are not “interaction” terms because the components are not independent.

Table 6: precise that non significant interactions (α=0.05?) were removed from the model.

Yes, nonsignificant terms were removed. Since the terms are nonsignificant they contribute little to the model. Terms can be removed from a model as long as hierarchy is maintained. Model hierarchy requires that all of the lower-order terms that make up any higher-order terms by included in the model. Removing nonsignificant terms provides 1 df for each term removed and this often results in better models.

L459-461: it seems that to make this conclusion we should have non-infested P. lunatus.

We will continue investigating whether there is a banker plant effect.

---

## [Decision Letter · Decision Letter 1]

4 Mar 2020

PONE-D-19-26873R1

Developing a conservation biological control strategy for the Asian citrus psyllid: Response surface methodology reveals proportionality effects of plant species in conservation plantings

PLOS ONE

Dear Dr. Patt,

Thank you for submitting your manuscript to PLOS ONE. After careful consideration, we feel that it has merit but does not fully meet PLOS ONE’s publication criteria as it currently stands. Therefore, we invite you to submit a revised version of the manuscript that addresses the points raised during the review process.

This manuscript presented a difficult decision. One reviewer previously suggested rejected, and upon review of the revisions retained that recommendation. In light of that, an additional, highly expert, review was requested. That reviewer suggested minor, but very important, revisions. They also agreed with some of the concerns previously expressed. In light of that, I am recommending that you carefully consider the comments and incorporate them into a revision.

We would appreciate receiving your revised manuscript by Apr 18 2020 11:59PM. To enhance the reproducibility of your results, we recommend that if applicable you deposit your laboratory protocols in protocols.io, where a protocol can be assigned its own identifier (DOI) such that it can be cited independently in the future. For instructions see: http://journals.plos.org/plosone/s/submission-guidelines#loc-laboratory-protocols

We look forward to receiving your revised manuscript.

Kind regards,

Sean Michael Prager, Ph.D.

Academic Editor

PLOS ONE

Reviewers' comments:

Reviewer's Responses to Questions

**Comments to the Author**

1. If the authors have adequately addressed your comments raised in a previous round of review and you feel that this manuscript is now acceptable for publication, you may indicate that here to bypass the “Comments to the Author” section, enter your conflict of interest statement in the “Confidential to Editor” section, and submit your "Accept" recommendation.

Reviewer #1: All comments have been addressed

Reviewer #4: (No Response)

2. Is the manuscript technically sound, and do the data support the conclusions?

Reviewer #1: Partly

Reviewer #4: Yes

3. Has the statistical analysis been performed appropriately and rigorously? 

Reviewer #1: Yes

Reviewer #4: Yes

4. Have the authors made all data underlying the findings in their manuscript fully available?

Reviewer #1: Yes

Reviewer #4: Yes

5. Is the manuscript presented in an intelligible fashion and written in standard English?

Reviewer #1: Yes

Reviewer #4: Yes

6. Review Comments to the Author

Reviewer #1: The authors did an excellent job at responding to comments. However, I do not think this a fit for this journal as is. While the design of the study is novel, it is a local, proof-of-concept study, as suggested by the authors themselves, and the yellow sticky card sampling method is not the most effective way to measure these communities in order to come to conclusions about effectiveness. From author comments, it seems like this will be a stronger paper submitted later with new sampling methods.

Reviewer #4: Patt et al. present a very interesting, microscale experiment testing the effect of plant mixtures on abundance of communities of predatory arthropods. An interesting experimental design was employed that allowed the investigators to draw inferences and make statistically valid conclusions on the effects of plant species mixtures on abundance of arthropod species trapped within those mixtures of plants. The sampling method employed consisted of yellow sticky traps, which ensnared insects that were presumably attracted to the various plant mixture treatments. The authors ultimate goal is to improve conservation biological control of Asian citrus psyllid, which is a pest of commercially farmed citrus. For commercial production, citrus is grown in large monocultures. This psyllid species is a mobile vector of a bacterial pathogen, which severely limits citrus production in areas where the disease becomes widespread.

The manuscript has been thoroughly reviewed by three previous reviewers and the authors have submitted a response where significant effort was made to address the reviewer concerns. In my opinion, the three reviewers insightfully brought up several shortcomings of the manuscript and constructively suggested improvements. I agree with reviewer 1, that although the investigation is most interesting, it is not relevant to Asian citrus psyllid and citrus production per se, at this initial stage. Understandably, this is a first step as the authors clearly state in both the manuscript and their rebuttal. However, the very title of the current manuscript is misleading. This investigation included neither the Asian citrus psyllid, nor biological control of this pest. Also, as the reviewer points out, it was not conducted in a habitat that would require modification for enhancing biological control of this pest, nor were any specialist natural enemies of Asian citrus psyllid investigated. The plant species investigated modified the community structure of generalist predators that could play a role in biological control of Asian citrus psyllid or any number of similar hemipteran pests. This is all eventually stated/explained in the manuscript and with a careful read, it becomes clear. I understand that the authors are building context for their future investigations and this is considered a first step. Therefore, it should not be an issue to fix the manuscript to make it less misleading by modifying the title and making this clear upfront in both the abstract and introduction. The title can be changed to reflect what was actually done in the manuscript by deleting the statement prior to the colon and simply re-titling the manuscript; “Response surface methodology reveals proportionality effects of plant species in conservation plantings on populations of generalist predatory arthropods.”

A methodological shortcoming of this investigation, also pointed out by the reviewers, was the use of yellow sticky traps as the sole means of collecting data. This was thoroughly discussed by the reviewers and I think the authors have made a good effort to address this in the revised version of their manuscript. I see no reason to discuss this further and I think it was adequately addressed.

It appears that the authors have several interesting investigations planned or in progress that should provide a useful follow up to this initial proof of concept work.

Minor edits: line 421: “Proof of concept” is not a valid justification for not knowing “why” in this case. In my opinion, it’s not important to even justify that this answer is not yet known here; but if so, simply state you don’t yet know “why”. The hypothesis tests have not yet been conducted and stating the hypotheses and possibly suggesting how they may be tested is sufficient here.

I suggest including the following citation in the literature survey, which should also help the authors justify why their research is timely and useful:

Monzó, C. and Stansly, P.A. (2020), Economic value of conservation biological control for management of the Asian citrus psyllid, vector of citrus Huanglongbing disease. Pest Manag Sci. doi:10.1002/ps.5691

7. PLOS authors have the option to publish the peer review history of their article (what does this mean?). If published, this will include your full peer review and any attached files.

Reviewer #1: No

Reviewer #4: No

---

## [Author Response · Author response to Decision Letter 1]

23 Mar 2020

Dear Dr. Prager:

I have made the corrections suggested by Reviewer 4, which included the title change. I wish to thank you for your patience and helpfulness in completing the review. I also wish to extend my thanks and gratitude to all of the reviewers; their comments and suggestions have greatly improved the manuscript.

Best Regards—Joe Patt

---

## [Editor Report · Decision Letter 2]

25 Mar 2020

Response surface methodology reveals proportionality effects of plant species in conservation plantings on occurrence of generalist predatory arthropods

PONE-D-19-26873R2

Dear Dr. Patt,

We are pleased to inform you that your manuscript has been judged scientifically suitable for publication and will be formally accepted for publication once it complies with all outstanding technical requirements.

With kind regards,

Sean Michael Prager, Ph.D.

Academic Editor

PLOS ONE
---

## [Editor Report · Acceptance letter]

26 Mar 2020

PONE-D-19-26873R2 

Response surface methodology reveals proportionality effects of plant species in conservation plantings on occurrence of generalist predatory arthropods 

Dear Dr. Patt:

I am pleased to inform you that your manuscript has been deemed suitable for publication in PLOS ONE. Congratulations! Your manuscript is now with our production department. 

With kind regards,

on behalf of

Dr. Sean Michael Prager 

Academic Editor

PLOS ONE